# Decoupling Global Structure and Local Refinement: Blueprint-Guided Scroll Generation with Direct Preference Optimization

## Abstract

Existing methods for generating long scroll images, often fail to maintain global structural and stylistic consistency, resulting in artifacts like content repetition. To address this, we propose the Dual-Resolution Scroll Generation with Preference Optimization (DRSPO) framework. Our approach decouples global composition from local refinement by first generating a low-resolution (LR) blueprint to establish a coherent overall structure. This LR blueprint then guides a high-resolution (HR) feature to render fine-grained details. We further enhance generation quality by incorporating Direct Preference Optimization (DPO) at both stages, and we introduce a novel theoretical adaptation to apply preference tuning directly to the region-based generation process. Experimental results demonstrate that our method produces high-quality long scroll images with reasonable global structure and fine-grained details.

## 1 Introduction

The generation of long scroll images presents a challenging task within the field of image generation. Currently, mainstream image generation models primarily focus on producing images with an aspect ratio of 1:1. Directly applying these models like SDXL (Podell et al., 2023) or Flux (Labs, 2024) to generate high aspect ratio images encounters the Out-of-Distribution (OOD) problem, leading to a significant degradation in generation quality.

A prevalent strategy for high-resolution long scroll generation involves region-based methods, epitomized by MultiDiffusion (Bar-Tal et al., 2023). This approach is particularly effective at rendering high-fidelity local details by processing the image in smaller, manageable regions. Building on this foundation, subsequent works have introduced targeted improvements: Merge-Attend-Diffuse (MAD) operator (Quattrini et al., 2024) enhances semantic coherence by merging features across different window views, SyncDiffusion (Lee et al., 2023) focuses on improving stylistic consistency between these windows, and ElasticDiffusion (Haji-Ali et al., 2024) further refines overall local coherence. Conversely, an alternative paradigm involves the direct end-to-end application of powerful base models like SDXL (Podell et al., 2023) and Flux (Labs, 2024). This approach demonstrates a distinct advantage in establishing global coherence and stylistic consistency across the entire scroll.

Despite these advancements, both paradigms suffer from some limitations. The core deficiency of region-based methods is the absence of a holistic global plan. This frequently manifests as severe artifacts, most notably the unnatural repetition of objects, as illustrated in Fig. 1. While the direct application of base models provides more robust global control, their ability to render fine-grained details does not match the fidelity achieved by region-based techniques. Consequently, existing works present a fundamental tradeoff between global structural coherence and fine-grained local refinement.

To address these challenges, we propose the Dual-Resolution Scroll Generation with Preference Optimization (DRSPO) framework. Our framework is comprised of two core components. The first is a Dual-Resolution Generation Pipeline (detailed in Sec.3.1), which utilizes a low-resolution (LR) model to generate a blueprint that establishes the global compositional structure. This LR blueprint subsequently provides robust guidance for the high-resolution (HR) generation stage. The second component is a novel Direct Preference Optimization (DPO) method tailored for the MultiDiffusion

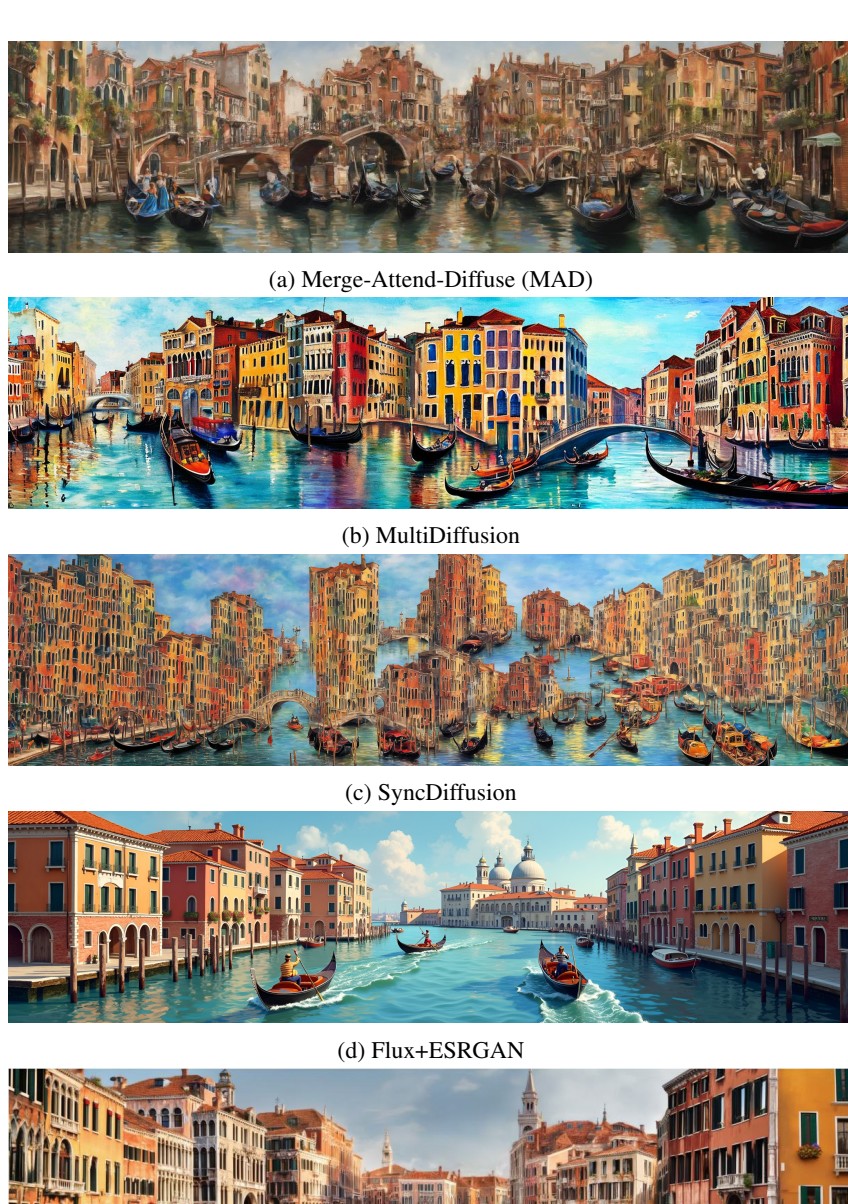

(a) Merge-Attend-Diffuse (MAD)

(b) MultiDiffusion

(c) SyncDiffusion

(d) Flux+ESRGAN

(e) DRSPO(Ours)

Figure 1: Scrolls(1024×4096) generated using prompts:"Paint a scene of a canal-laced city, like Venice or Amsterdam. Gondolas or boats glide through the waterways, passing under arched bridges and alongside colorful, historic houses that seem to rise directly from the water." As can be observed, the images generated by our method exhibit more reasonable and coherent layouts. Flux+ESRGAN also generates well-defined results. In contrast, other methods, which lack guidance from global information, produce images with severe object repetition.

process(detailed in Sec.3.2). It directly fine-tunes the base generator to enhance the fine-grained details within each local region. The efficacy of our framework is demonstrated by state-of-the-art performance on metrics such as HPS v2 and ImageReward. Visually, Fig. 1 confirms that our method excels at synthesizing coherent global structures with high-fidelity local details.

Our work introduces a Dual-Resolution Scroll Generation Method that enables controlled generation of the global structure, ensuring the final long scroll possesses both a coherent global composition

and consistent style. Furthermore, we adapt the DPO method for application within the MultiDiffusion scenario, specifically to optimize the generation of long scroll images. The effectiveness of this combined approach is demonstrated by our experimental results, which show our method's ability to generate high-quality outputs and achieve leading scores on key metrics for human preference and semantic fidelity.

## 2 PRELIMINARY

### 2.1 MULTIDIFFUSION

MultiDiffusion (Bar-Tal et al., 2023) is a training-free method that adapts a pre-trained diffusion model for long scroll generation using a sliding window approach. It is built upon a base model $\Phi$ whose standard denoising step is formulated as:

$$\Phi : \mathcal{I} \times \mathcal{Y} \to \mathcal{I}, \mathcal{I} \in \mathbb{R}^{H \times W \times C} \tag{1}$$

where $\Phi$ maps a noisy latent from the image space $\mathcal{I}$ to a denoised one under conditions from $\mathcal{Y}$, such that $I_{t-1} = \Phi(I_t|y)$. To generate a long scroll image $\mathcal{J} \in \mathbb{R}^{H' \times W' \times C}$, MultiDiffusion defines a new process $\Psi$:

$$\Psi : \mathcal{J} \times \mathcal{Z} \to \mathcal{J} \tag{2}$$

The new diffusion step, $J_{t-1} = \Psi(J_t|y)$, is constrained by the base model $\Phi$. This link is established via a mapping $F_i$ that extracts the $i$-th window and its inverse $F_i^{-1}$ that maps it back. The objective is to align this new process with the pre-trained model by solving the following optimization at each step:

$$\Psi(J_t|y) = \arg\min_{J \in \mathcal{J}} \sum_{i=1}^{n} \|F_i(J) - \Phi(F_i(J_t), y_i)\| \tag{3}$$

In practice, this optimization is implemented by denoising each window individually and then composing the complete long scroll $J_{t-1}$ via a weighted average of the results:

$$J_{t-1} = \frac{\sum_i F_i^{-1}(W_i \Phi(F_i(J_t), y_i))}{\sum_i F_i^{-1}(W_i)} \tag{4}$$

### 2.1.1 DPO-DIFFUSION

Direct Preference Optimization (DPO) (Rafailov et al., 2023; Wallace et al., 2024) is a method that fine-tunes a model to align with preferences by directly optimizing a policy against a reward function implicitly defined by a preference dataset. In the context of diffusion models, DPO-Diffusion defines this reward over the entire denoising trajectory:

$$r(c, x_0) = \mathbb{E}_{p_\theta(x_{1:T}|x_0,c)}[R(c, x_{0:T})] \tag{5}$$

The reinforcement learning (RL) objective, using the KL divergence as a constraint, is formulated as:

$$\max_{p_\theta} \mathbb{E}_{c \sim \mathcal{D}_c, x_{0:T} \sim p_\theta(x_{0:T}|c)}[r(c, x_0)] - \beta \mathbb{D}_{KL}[p_\theta(x_{0:T}|c) \| p_{\text{ref}}(x_{0:T}|c)] \tag{6}$$

By substituting the difference in rewards for preferred $(x_0^w)$ and dispreferred $(x_0^l)$ samples from a preference dataset $\mathcal{D}$, the final DPO-Diffusion loss function is derived as:

$$
\begin{aligned}
L_{\text{DPO-Diffusion}} = -\mathbb{E}_{(c,x_0^w,x_0^l) \sim \mathcal{D}} \log \sigma \Big( \beta \mathbb{E}_{x_{1:T}^w \sim p_\theta(\cdot|x_0^w,c), x_{1:T}^l \sim p_\theta(\cdot|x_0^l,c)} \\
\Big[ \log \frac{p_\theta(x_{0:T}^w|c)}{p_{\text{ref}}(x_{0:T}^w|c)} - \log \frac{p_\theta(x_{0:T}^l|c)}{p_{\text{ref}}(x_{0:T}^l|c)} \Big] \Big)
\end{aligned} \tag{7}
$$

## 3 METHOD

Existing high-resolution long scroll generation methods face challenges with global structure and content repetition. Approaches like MultiDiffusion (Bar-Tal et al., 2023), which rely on "divide and conquer" strategy of independently denoising and then aggregating image regions, inherently

lack the global context necessary for structural coherence. While the end-to-end methods lack local refinement. These limitations highlight the need for an external control mechanism to impose global structural consistency while maintaining the local details.

To address these issues, we propose the Dual-Resolution Scroll Generation with Preference Optimization (DRSPO) framework. Our method first uses a Low-Resolution (LR) generator to establish a coherent global blueprint, which then guides a High-Resolution (HR) generator to render fine-grained details. We further enhance this pipeline by integrating DPO (Wallace et al., 2024) at both stages, fine-tuning the models on preference data to improve generation quality, resulting in scrolls that are both globally coherent and locally detailed.

## 3.1 GENERATION PIPELINE

Our framework is implemented as a two-stage generation pipeline: the Low-Resolution blueprint generation stage and the subsequent High-Resolution detail rendering stage. The framework of our generation pipeline is shown in Fig. 2.

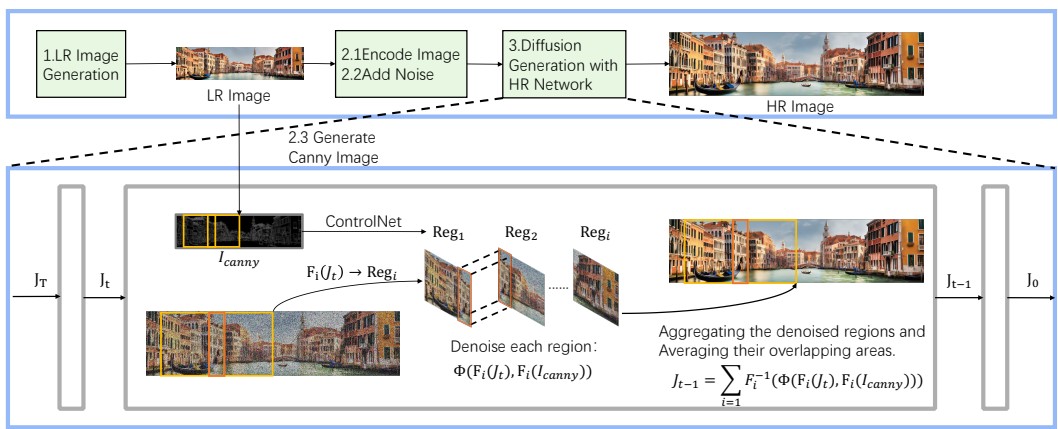

Figure 2: Detailed schematic of our Dual-Resolution Generation Pipeline. The process begins with the generation of a Low-Resolution (LR) image (Step 1), which serves as a global 'blueprint' for HR generation (Step 2.1-2.3). The lower panel provides an in-depth view of a single denoising step within the HR generator (Step 3).

### 3.1.1 LOW-RESOLUTION IMAGE GENERATION

The primary objective of this stage is to establish the global structure and style for the final long scroll. To circumvent the content repetition and structural control issues inherent in direct high-resolution synthesis, we first employ a pre-trained base model to generate the complete long scroll at a lower resolution. This process can be formalized as:

$$\Phi_{\text{LR}} : \mathcal{I} \times \mathcal{Y} \to \mathcal{I}, \mathcal{I} \in \mathbb{R}^{H_1 \times W_1 \times C}, I_{\text{LR}} = \Phi_{\text{LR}}(z, c) \tag{8}$$

where $I_{\text{LR}} \in \mathbb{R}^{H_1 \times W_1 \times c}$ is the generated LR image, and $\Phi_{\text{LR}}$ is the pre-trained model.

To further enhance the quality and consistency of the LR image, we fine-tune the model $\Phi_{\text{LR}}$ using DPO (Wallace et al., 2024). We construct a preference dataset by evaluating a corpus of long scroll images with an Aesthetic Score (Schuhmann et al., 2022), from which we form preference pairs. The detailed methodology for dataset construction is described in Sec. 5.1.1. The DPO fine-tuning process is guided by the following loss function:

$$L(\theta) = -\mathbb{E}_{t, J_t^w \sim q(J_t | J_0^w), J_t^l \sim q(J_t | J_0^l)} \log \sigma \Big( (-\beta T \omega(\lambda_t)) ($$

$$(\|\epsilon_t^w - \epsilon_\theta^w\| - \|\epsilon_t^w - \epsilon_{\text{ref}}^w\|) - (\|\epsilon_t^l - \epsilon_\theta^l\| - \|\epsilon_t^l - \epsilon_{\text{ref}}^l\|)) \Big) \tag{9}$$

### 3.1.2 HIGH-RESOLUTION SCROLL GENERATION

The High-Resolution generation stage employs a modified MultiDiffusion framework, where the Low-Resolution ($I_{\text{LR}}$) image from the preceding stage serves as a multi-source guidance signal. This approach ensures the final long scroll is rich in detail while strictly adhering to the established global structure.

Rather than using standard Gaussian noise, we initialize the process with the latent features of the LR image, obtained via an image encoder $\mathbf{E}$:

$$z_{\text{init}} = \mathbf{E}(I_{\text{LR}}), \, J = \Psi(z_{\text{init}}, \mathbf{c}) \tag{10}$$

In this formulation, $J \in \mathbb{R}^{H_2 \times W_2 \times c}$ is the final generated high-resolution long scroll. To ensure structural alignment, we maintain an identical aspect ratio $k$ for both HR and LR images, such that $\frac{W_1}{H_1} = \frac{W_2}{H_2} = k$.

**ControlNet** To impose structural control, the LR image is first converted into a line art representation, which guides the HR generation. Within the MultiDiffusion process, where each window is generated independently, ControlNet (Zhang et al., 2023) provides localized control for each corresponding region, as denoted by the mapping function $F_i, i \in \{1, \ldots, n\}$.

$$
\begin{aligned}
I_{\text{canny}} &= \text{Canny}(I_{\text{LR}}) \\
I_{t-1}^i &= \Phi(F_i(J_t), F_i(I_{\text{canny}}), \mathbf{c})
\end{aligned} \tag{11}
$$

## 3.2 APPLYING DPO TO MULTIDIFFUSION

In MultiDiffusion (Bar-Tal et al., 2023), each denoising step involves local denoising followed by inter-region fusion. Consequently, deriving the noisy latent $J_t$ at an arbitrary timestep $t$ from the scroll $J_0$ is non-trivial. To align with the DPO-Diffusion (Wallace et al., 2024) derivation, we approximate the forward process $q(J_t|J_0)$ by applying noise directly to the entire image (Ho et al., 2020; Song et al., 2020):

$$q(J_t|J_0) = \mathcal{N}(J_t; \sqrt{\alpha_t}J_0, (1 - \alpha_t)\mathbf{I}) \tag{12}$$

We then define the reward function on the complete long scroll image:

$$r(c, J_0) = \mathbb{E}_{p_\theta(J_{1:T}|J_0, c)}[R(c, J_{0:T})] \tag{13}$$

Maximizing this reward while constraining the policy via KL divergence gives:

$$\max_{p_\theta} \mathbb{E}_{c \sim \mathcal{D}_c, J_{0:T} \sim p_\theta(J_{0:T}|c)}[r(c, J_0)] - \beta \mathbb{D}_{KL}[p_\theta(J_{0:T}|c) \, || \, p_{\text{ref}}(J_{0:T}|c)] \tag{14}$$

Solving for the reward function yields:

$$
\begin{aligned}
R(c, J_{0:T}) &= \beta \log \frac{p_\theta^*(J_{0:T}|c)}{p_{\text{ref}}(J_{0:T}|c)} + \beta \log Z(c) \\
r(c, J_0) &= \beta \mathbb{E}_{p_\theta(J_{1:T}|J_0, c)}\left[\log \frac{p_\theta^*(J_{0:T}|c)}{p_{\text{ref}}(J_{0:T}|c)}\right] + \beta \log Z(c)
\end{aligned} \tag{15}
$$

The DPO loss function for long scroll generation is thus:

$$L_{\text{DPO-MultiDiffusion}} = -\mathbb{E}_{(c, J_0^w, J_0^l) \sim \mathcal{D}} \log \sigma$$

$$\left(\beta \mathbb{E}_{J_{1:T}^w \sim p_\theta(J_{1:T}|J_0^w, c), J_{1:T}^l \sim p_\theta(J_{1:T}|J_0^l, c)}\left[\log \frac{p_\theta(J_{0:T}^w|c)}{p_{\text{ref}}(J_{0:T}^w|c)} - \log \frac{p_\theta(J_{0:T}^l|c)}{p_{\text{ref}}(J_{0:T}^l|c)}\right]\right) \tag{16}$$

Since the reverse trajectory $p_\theta(x_{1:T}|x_0)$ is intractable, we follow the approach in DPO and substitute it with the forward process $q(x_{1:T}|x_0)$, using the simplified noising process defined in Equation. 12. By applying Jensen's inequality, we can derive the following upper bound:

$$
\begin{aligned}
L_{\text{DPO-MultiDiffusion}} \leq -\mathbb{E}_{t, J_t^w \sim q(J_t|J_0^w), J_t^l \sim q(J_t|J_0^l)} \log \sigma\Big(-\beta T\Big( \\
+ \mathbb{D}_{\text{KL}}\big(q(J_{t-1}^w|J_0^w, J_t^w) \, || \, p_\theta(J_{t-1}^w|J_t^w, c)\big) \\
- \mathbb{D}_{\text{KL}}\big(q(J_{t-1}^w|J_0^w, J_t^w) \, || \, p_{\text{ref}}(J_{t-1}^w|J_t^w, c)\big) \\
- \mathbb{D}_{\text{KL}}\big(q(J_{t-1}^l|J_0^l, J_t^l) \, || \, p_\theta(J_{t-1}^l|J_t^l, c)\big) \\
+ \mathbb{D}_{\text{KL}}\big(q(J_{t-1}^l|J_0^l, J_t^l) \, || \, p_{\text{ref}}(J_{t-1}^l|J_t^l, c)\big)\Big)\Big)
\end{aligned} \tag{17}
$$

In the MultiDiffusion framework, the mean of the denoised scroll is an aggregation of the means from each local window:

$$p(J_{t-1}|J_t, c) \approx \mathcal{N}\left(J_{t-1}; \frac{1}{n}\sum_{i=1}^{n} F_i^{-1}(\mu_{\theta,i}(J_t, t, c)), \Sigma_t\right) \tag{18}$$

where $\mu_{\theta,i}(J_t, t, c)$ is the mean predicted by the local denoiser $\Phi$ for window $i$. The loss can be expressed in terms of the predicted noise $\epsilon$ yields:

$$L(\theta) = -\mathbb{E}_{t, J_t^w \sim q(J_t|J_0^w), J_t^l \sim q(J_t|J_0^l)} \log \sigma\Bigg((-\beta T \omega(\lambda_t))\Big($$

$$\|\epsilon^w - \sum_{i=1}^{n} F_i^{-1}(\epsilon_\theta(x_t^w, t))\| - \|\epsilon^w - \sum_{i=1}^{n} F_i^{-1}(\epsilon_{\text{ref}}(x_t^w, t))\| \tag{19}$$

$$-(\|\epsilon^l - \sum_{i=1}^{n} F_i^{-1}(\epsilon_\theta(x_t^l, t))\| - \|\epsilon^l - \sum_{i=1}^{n} F_i^{-1}(\epsilon_{\text{ref}}(x_t^l, t))\|))\Bigg)$$

where $x_t^w = F_i(J_t^w)$ and $x_t^l = F_i(J_t^l)$ are the noisy latents for the respective windows.

### 3.3 DISCUSSION

#### 3.3.1 VARIANCE CONSIDERATIONS

In the derivation of Equation. 18, our analysis simplified the process by disregarding the effect of local window aggregation on the variance of the complete long scroll. However, as discussed in (Sun et al., 2025), the window stitching in MultiDiffusion has a non-trivial impact on variance. Assuming N windows overlap in a given region, the mean and variance of the denoised latent $x_{t-1}$ in that region are:

$$x_{t-1} \sim \mathcal{N}\left(\frac{\sum_i^N \mu_{t,i}}{N}, \frac{\sigma_t^2}{N}\right) \tag{20}$$

We now re-examine the derivation from Equation. 17 under this condition. The original derivation was based on the KL divergence between two Gaussian distributions with identical covariance $\Sigma = \sigma^2 \mathbf{I}$:

$$D_{\text{KL}}(\mathcal{N}(\boldsymbol{\mu}_1, \boldsymbol{\Sigma})\|\mathcal{N}(\boldsymbol{\mu}_2, \boldsymbol{\Sigma}))$$

$$= \frac{1}{2}\left(\text{tr}(\boldsymbol{\Sigma}^{-1}\boldsymbol{\Sigma}) + (\boldsymbol{\mu}_2 - \boldsymbol{\mu}_1)^T \boldsymbol{\Sigma}^{-1}(\boldsymbol{\mu}_2 - \boldsymbol{\mu}_1) - d + \log\frac{\det\boldsymbol{\Sigma}}{\det\boldsymbol{\Sigma}}\right) = \frac{1}{2\sigma^2}\|\boldsymbol{\mu}_1 - \boldsymbol{\mu}_2\|_2^2 \tag{21}$$

In the MultiDiffusion context, however, the effective variance is scaled by $1/N$. The KL divergence between two Gaussians (Hershey & Olsen, 2007) with differing variances ($\sigma_1^2$ and $\sigma_2^2$) is therefore more appropriate:

$$D_{\text{KL}}(P_1\|P_2) = \frac{1}{2}\left[2d\log\frac{\sigma_2}{\sigma_1} - d + d\frac{\sigma_1^2}{\sigma_2^2} + \frac{1}{\sigma_2^2}\|\boldsymbol{\mu}_1 - \boldsymbol{\mu}_2\|_2^2\right]$$

$$= \frac{1}{2\sigma_2^2}\|\boldsymbol{\mu}_1 - \boldsymbol{\mu}_2\|_2^2 + \frac{1}{2}\left(2d\log\frac{\sigma_2}{\sigma_1} - d + \frac{d\sigma_1^2}{\sigma_2^2}\right) \tag{22}$$

Crucially, since the ratio of the variances $\frac{\sigma_2}{\sigma_1}$ is a constant in the MultiDiffusion scenario, this more accurate KL divergence formulation differs from the original only by an additive constant term. A term that is independent of the model parameters $\theta$ does not affect the location of the optima. Therefore, this variance consideration does not alter the DPO optimization process, and our previously derived loss function remains valid.

#### 3.3.2 LIMITATIONS

To apply the DPO framework to the MultiDiffusion process, we made a simplifying assumption about the forward noising process (Equation. 12), which may not perfectly align with the true dynamics of MultiDiffusion's region-based aggregation.

Furthermore, our data selection strategy, which constructs preference pairs by comparing the holistic quality of entire long scrolls, could be refined. The overall quality of a long scroll does not always correlate with the quality of its constituent regions; for example, a globally dispreferred image may still contain locally high-quality regions. A promising direction for future work would be to adopt a hybrid training strategy that simultaneously applies DPO at both the global long scroll level and the separate region level. This could be implemented via a composite loss function:

$$L = L_1(\theta) + \lambda L_2(\theta) \tag{23}$$

where $L_1(\theta)$ is the long scroll DPO loss from Equation. 19, and $L_2(\theta)$ is a DPO loss function trained on individual image regions. The $\lambda$ would balance two objectives.

## 4 RELATED WORK

### 4.1 HIGH-ASPECT-RATIO IMAGE GENERATION

Standard text-to-image diffusion models, primarily trained on square images, struggle to generate high-aspect-ratio content, often producing repetitive or incoherent structures when tasked with out-of-distribution sizes. To overcome this, region-based, sliding-window approaches have become the dominant paradigm. MultiDiffusion (Bar-Tal et al., 2023)pioneered this area by denoising overlapping regions on a large canvas and blending them to ensure local coherence.

Building on this foundation, subsequent research has aimed to mitigate the artifacts inherent in this local-to-global process. For instance, Merge-Attend-Diffuse (MAD) operator (Quattrini et al., 2024) enhances long-range semantic consistency by sharing features across non-adjacent windows, while SyncDiffusion (Lee et al., 2023) and ElasticDiffusion (Haji-Ali et al., 2024) focus on improving stylistic uniformity and transitional smoothness. More recent methods like TwinDiffusion (Zhou & Tang, 2024) and SpotDiffusion (Frolov et al., 2024) have sought to further refine quality and optimize the efficiency of this region-based pipeline. Native high-resolution generators, such as the Transformer-based PixArt-$\alpha$ (Chen et al., 2023) and PixArt-$\Sigma$(Chen et al., 2024), leverage efficient DiT architectures and high-quality 4K training data to directly synthesize high-fidelity images. Alternatively, cascaded pipelines employ a "generate-then-upscale" strategy, utilizing foundation models (e.g., Stable Diffusion) followed by super-resolution networks like Real-ESRGAN (Wang et al., 2021) to enhance resolution. However, despite these incremental improvements, all such methods lack a dedicated mechanism for high-level compositional planning. This fundamental gap often leads to images that are locally seamless but globally repetitive or structurally flawed. Our work directly addresses this by introducing a global planning stage via a dual-resolution framework.

### 4.2 PREFERENCE OPTIMIZATION IN GENERATIVE MODELS

Aligning generative models with human preferences, such as aesthetic quality and prompt fidelity, is a critical challenge. Reinforcement Learning from Human Feedback (RLHF) (Ouyang et al., 2022) was a foundational approach, where a separate reward model is trained on human preference data to guide the generator. However, RLHF is often complex and unstable to train. A more direct and stable alternative, DPO (Rafailov et al., 2023), has recently gained prominence. DPO bypasses the need for an explicit reward model by reframing the objective as a simple classification loss on preferred and dispreferred samples, fine-tuning the policy directly. This technique has been successfully adapted for diffusion models in works like Diffusion-DPO (Wallace et al., 2024), proving effective at enhancing image quality and alignment. In our framework, we apply DPO not only to enhance global structural control during the planning stage but also extend the method itself to optimize the region-based MultiDiffusion process.

## 5 EXPRIMENTS

### 5.1 SETTINGS

#### 5.1.1 PREFERENCE DATASET CONSTRUCTION

To facilitate Direct Preference Optimization (DPO) (Wallace et al., 2024), we constructed a large-scale preference dataset for long scroll images. The foundation of this dataset is a curated set of

4,936 high-quality prompts, which were derived from the corpus of (Zhang et al., 2024a). We use Coze to build a LLM-based workflow that filtered for outdoor scenes and textually augmented the content. Following the methodology of (Zhang et al., 2024b) to ensure sample diversity, we generated a comparison group of six distinct images for each prompt using three models (MultiDiffusion (Bar-Tal et al., 2023), Stable Diffusion XL (Podell et al., 2023), and MAD (Quattrini et al., 2024)) with two random seeds. Subsequently, every generated image was evaluated using aesthetic score (Schuhmann et al., 2022). By ranking the images within each comparison group based on these scores, we systematically established preference hierarchies, allowing us to extract the "preferred" and "dispreferred" sample pairs essential for aligning our models with desired aesthetic and structural qualities through DPO training.

### 5.1.2 HYPERPARAMETERS

For DPO training, we use Adafactor to save memory. We train our model on single NVIDIA H800 GPU using batch size of 1 pair. We train at resolution $1024 \times 4096$ for both stages. And for HR model, we use stride=32 and window size=128, corresponding to the resolution $1024 \times 4096$. For both stage, we use Low rank adaptation(LoRA) to save memory and the rank is set to 64. We use learning rate=1e-6 with 1000 warm up steps. Following DPO, we use $\beta = 5000$ during training.

### 5.1.3 EVALUATION

For our evaluation, we constructed a test set from two sources: 220 new prompts for outdoor scenes generated by Google Gemini, and a hold-out set of 200 prompts from our training distribution that were not used during training. We then employed several long scroll generation methods to produce images for each prompt, benchmarking our approach against prominent baseline methods including MultiDiffusion (Bar-Tal et al., 2023), SDXL (Podell et al., 2023), MAD (Quattrini et al., 2024), ElasticDiffusion (Haji-Ali et al., 2024), SyncDiffusion (Lee et al., 2023), and the FLUX.1-dev (Labs, 2024) version. The resulting images were assessed using a suite of quantitative metrics. To evaluate overall image quality and aesthetic appeal, we utilized three established scoring models: Aesthetic Score (Schuhmann et al., 2022), HPS v2 (Wu et al., 2023),ImageReward (Kirstain et al., 2023) and PickScore (Kirstain et al., 2023). We used the CLIP score (Hessel et al., 2021) to measure the semantic consistency between the generated images and their corresponding prompts.

### 5.2 QUANTITATIVE RESULTS

Table 1: Quantitative results of scrolls generated by different models. The numbers in parentheses indicate the specific rank within the group (lower is better). The 1st , 2nd and 3rd best values are highlighted with background colors. The **Average Rank** column reflects the overall balanced performance.

| Model | Aesthetic Score ↑ | HPS v2 ↑ | PickScore ↑ | ImageReward ↑ | CLIP Score ↑ | Average Rank ↓ |
|---|---|---|---|---|---|---|
| MultiDiffusion | 5.616 (8) | 0.226 (4) | 0.070 (7) | 0.224 (5) | 0.317 (2) | 5.2 |
| SdXL | 5.921 (4) | 0.190 (8) | 0.076 (6) | -0.318 (8) | 0.310 (4) | 6.0 |
| FLUX | 4.994 (10) | 0.175 (10) | 0.027 (9) | -1.084 (9) | 0.270 (10) | 9.6 |
| MAD | 6.032 (1) | 0.195 (7) | 0.105 (1) | 0.100 (6) | 0.317 (2) | 3.4 |
| SyncDiffusion | 5.744 (6) | 0.197 (6) | 0.068 (8) | -0.087 (7) | 0.310 (4) | 6.2 |
| ElasticDiffusion | 5.422 (9) | 0.186 (9) | 0.025 (10) | -1.351 (10) | 0.273 (9) | 9.4 |
| FLUX + ESRGAN | 5.922 (3) | 0.235 (2) | 0.097 (2) | 0.236 (4) | 0.306 (8) | 3.8 |
| SD 3.5 + ESRGAN | 5.759 (5) | 0.221 (5) | 0.092 (4) | 0.258 (3) | 0.309 (6) | 4.6 |
| PixArt | 6.016 (2) | 0.236 (1) | 0.092 (5) | 0.397 (1) | 0.308 (7) | 3.2 |
| DRSPR (Ours) | 5.726 (7) | 0.228 (3) | 0.096 (3) | 0.275 (2) | 0.321 (1) | 3.2 |

The quantitative results clearly demonstrate the superiority of our proposed method. Our model significantly outperforms competing approaches on the metrics most aligned with image quality preference and semantic fidelity, achieving the highest scores in HPS v2, ImageReward, and CLIP Score, along with second-place in PickScore. This robust performance validates the effectiveness of our DPO-based pipeline in producing visually appealing, compositionally sound, and semantically coherent images. Notably, while our method does not achieve the top rank on the Aesthetic Score, we attribute this to the metric's well-documented bias towards favoring high-frequency local textures and details over global structural coherence.

## 5.3 QUALITATIVE RESULTS

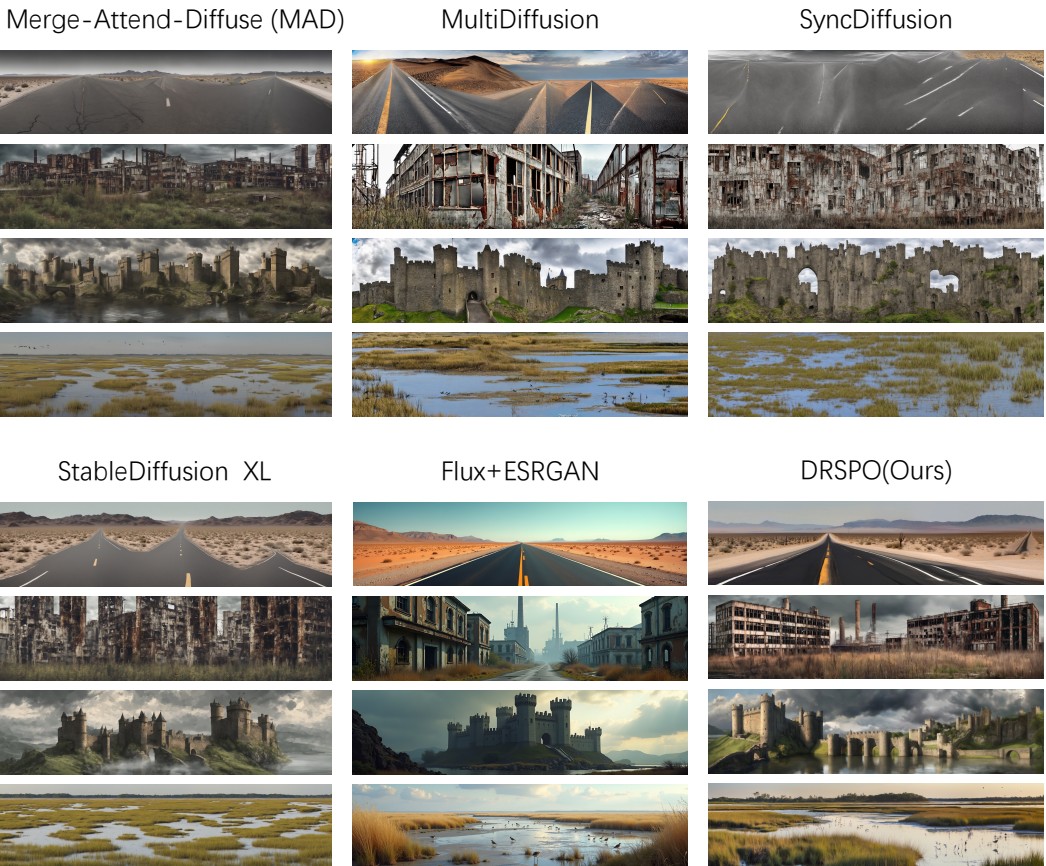

Figure 3: Further qualitative comparisons between our method and other baselines.

We provide more comparisons with other baselines as shown in Fig. 3. Baseline methods frequently suffer from object repetition. For instance, in the first row, other approaches generate multiple roads, disrupting the scene's logical coherence. Our method successfully eliminates the repetitive artifacts that plague competing approaches, achieving better global structural coherence and fine-grained details.

## 5.4 ABLATION STUDY

We conduct an ablation study to validate the contributions of our key components: Low-Resolution blueprint guidance and DPO training for high-resolution generation. The results in Table. 2 reveal their distinct roles. The removal of either component leads to a notable degradation in the final output. Specifically, omitting DPO training primarily harms human preference scores, whereas omitting the LR guidance causes a sharp HPS v2 score decline. This confirms their complementary roles. While the results of removing control blocks and DPO blocks both score higher on the Aesthetic Score, we attribute this to the metric's known bias towards local textures, which fails to penalize its global incoherence. Furthermore, integrating an additional control mechanism (IP-Adapter) proves detrimental, degrading performance across all metrics.

## 6 CONCLUSION

In this work, we propose the Dual-Resolution Scroll Generation with Preference Optimization (DR-SPO) framework. Our method decouples global composition from local refinement by using a low-resolution blueprint to guide a region-based high-resolution model. A key contribution is our novel

Table 2: Ablation study on our key components. The numbers in parentheses indicate the specific rank within the group (lower is better). The 1st , 2nd and 3rd best values are highlighted with background colors. The **Average Rank** column reflects the overall balanced performance.

| Model | Aesthetic Score ↑ | HPS v2 ↑ | PickScore ↑ | ImageReward ↑ | CLIP Score ↑ | Average Rank ↓ |
|---|---|---|---|---|---|---|
| DRSPO(Ours) | 5.726 (3) | 0.228 (1) | 0.134 (2) | 0.275 (1) | 0.321 (1) | 1.6 |
| w/o control | 6.009 (1) | 0.136 (4) | 0.166 (1) | 0.171 (3) | 0.320 (2) | 2.2 |
| w/o DPO | 5.734 (2) | 0.220 (2) | 0.129 (3) | 0.247 (2) | 0.320 (2) | 2.2 |
| w IP-Adapter | 5.564 (4) | 0.216 (3) | 0.086 (4) | -0.058 (4) | 0.313 (4) | 3.8 |

adaptation of Direct Preference Optimization (DPO) to the MultiDiffusion process, which we apply at both stages to align the output with human quality preferences. Experimental results validate our approach, demonstrating a state-of-the-art balance of global structural integrity and fine-grained details. Future work in long-scroll generation will continue to focus on the central challenge of simultaneously achieving coherent global structures and rendering fine-grained, high-fidelity details.

## 7 THE USE OF LARGE LANGUAGE MODELS (LLMs)

For the preparation of this manuscript, we utilized Large Language Model (LLM) based tools to assist with improving the language, clarity, and readability of the prose. We wish to clarify that the application of these tools was strictly confined to stylistic and grammatical refinement. The core scientific contributions, including the conceptual framework, methodology, and experimental analysis, were developed entirely by the authors.

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

# A  APPENDIX

## A.1  INTRODUCTION ON DIRECT PREFERENCE OPTIMIZATION

**DPO**  The RLHF (Reinforcement Learning from Human Feedback) pipeline comprises three primary stages:

1. Supervised Fine-tuning (SFT): Adapts a pre-trained large language model to downstream tasks.

2. Preference Sampling and Reward Learning: - Generates data pairs using the SFT model: $(y_1, y_2) = \pi^{SFT}(y \mid x)$ - Human evaluation yields preference relations $y_w \succ y_l \mid x$. - A theoretically optimal reward model is denoted $r^*(x, y)$. - Utilizing the Bradley-Terry (BT) model, the user preference distribution is:

$$p^*(y_1 \succ y_2 | x) = \frac{\exp(r^*(x, y_1))}{\exp(r^*(x, y_1)) + \exp(r^*(x, y_2))} = \sigma(r^*(x, y_1) - r^*(x, y_2)) \qquad (24)$$

- Given a dataset $\mathcal{D} = \{x^{(i)}, y_w^{(i)}, y_l^{(i)}\}_{i=1}^N$ sampled from $p^*$, the reward model $r_\phi$ is optimized via maximum likelihood estimation:

$$\mathcal{L}_R(r_\phi, \mathcal{D}) = -\mathbb{E}_{(x, y_w, y_l) \sim \mathcal{D}}[\log \sigma(r_\phi(x, y_w) - r_\phi(x, y_l))] \qquad (25)$$

3. RL Optimization: Fine-tunes the policy using the learned reward model:

$$\max_{\pi_\theta} \mathbb{E}_{x \sim \mathcal{D}, y \sim \pi_\theta(y|x)}[r_\phi(x, y)] - \beta \mathbb{D}_{KL}[\pi_\theta(y \mid x) \mid\mid \pi_{\text{ref}}(y \mid x)] \qquad (26)$$

Or equivalently in a diffusion context:

$$\max_{p_\theta} \mathbb{E}_{c \sim \mathcal{D}_c, x_{0:T} \sim p_\theta(x_{0:T}|c)}[r(c, x_0)] - \beta \mathbb{D}_{KL}[p_\theta(x_0|c) \mid\mid p_{\text{ref}}(x_0|c)] \qquad (27)$$

DPO optimizes RLHF by analytically deriving the reward model expression, simplifying training. The optimal policy under the reward $r$ is:

$$\pi_r(y \mid x) = \frac{1}{Z(x)} \pi_{\text{ref}}(y \mid x) \exp\left(\frac{1}{\beta} r(x, y)\right) \qquad (28)$$

where $Z(x) = \sum_y \pi_{\text{ref}}(y \mid x) \exp\left(\frac{1}{\beta} r(x, y)\right)$ is the partition function. This represents a valid probability distribution. Solving for the reward function yields:

$$r(x, y) = \beta \log \frac{\pi_r(y \mid x)}{\pi_{\text{ref}}(y \mid x)} + \beta \log Z(x) \qquad (29)$$

Substituting this into the BT model preference probability gives:

$$p^*(y_1 \succ y_2 | x) = \frac{1}{1 + \exp\left(\beta \log \frac{\pi_r(y_2|x)}{\pi_{\text{ref}}(y_2|x)} - \beta \log \frac{\pi_r(y_1|x)}{\pi_{\text{ref}}(y_1|x)}\right)} \qquad (30)$$

The DPO loss function is the negative log-likelihood of this probability:

$$\mathcal{L}_{DPO}(\pi_\theta; \pi_{\text{ref}}) = -\mathbb{E}_{(x, y_w, y_l) \sim \mathcal{D}}\left[\log \sigma\left(\beta\left(\log \frac{\pi_\theta(y_w \mid x)}{\pi_{\text{ref}}(y_w \mid x)} - \log \frac{\pi_\theta(y_l \mid x)}{\pi_{\text{ref}}(y_l \mid x)}\right)\right)\right] \qquad (31)$$

## A.2  OTHER CONTROL METHOD FOR HIGH-RESOLUTION SCROLL GENERATION

In Sec. 3.1.2, we introduce the process of High-Resolution Scroll Generation. We also tried other method to control the global structure using low-resolution blueprint.

**IP-Adapter**  We inject the $I_{\text{LR}}$ image as an image prompt into the MultiDiffusion process. A pretrained image encoder $\mathcal{E}$ is used to encode $I_{\text{LR}}$:

$$\mathbf{F}(I_{\text{LR}}) = \mathcal{E}(I_{\text{LR}}) \qquad (32)$$

Subsequently, a lightweight adapter network projects these features to obtain the image conditioning vector, $c_{\text{image}} = \mathcal{P}(\mathbf{F}(I_{\text{LR}}))$. The final U-Net conditioning is a concatenation of text and image prompts: $\mathbf{c} = [\mathbf{c}_{\text{text}}, \mathbf{c}_{\text{image}}]$. This design is highly parameter-efficient and enables flexible, multi-modal control. Equation 11 is thus modified to:

$$I_{t-1}^i = \Phi(F_i(J_t), F_i(I_{\text{canny}}), [\mathbf{c}_{\text{text}}, \mathbf{c}_{\text{image}}]) \tag{33}$$

We ablate this block in our ablation study detailed in Sec. 5.1.1, where the results show that incorporating the Ip-Adapter block into our method degrades performance. We also find that this block introduces a blurring effect, which adversely impacts the overall image quality. Moreover, an excessive number of control modules can cause the generated image to adhere too rigidly to the control blueprint, thereby preventing the achievement of our desired creative outcome.

### A.3 DPO FOR MULTIDIFFUSION

In this section, we will introduce the details of our proposed MultiDiffusion-DPO method. The base loss function, presented in Equation. 16, is difficult to optimize directly because the reverse process trajectory $p_\theta(x_{1:T}|x_0)$ is intractable. To overcome this, we adopt an approximation strategy similar to that used in the original DPO paper. We substitute the intractable reverse process with the tractable forward noising process, $q(x_{1:T}|x_0)$. This substitution, followed by the application of Jensen's inequality, allows us to derive a final, optimizable loss function.

$$
\begin{aligned}
L_{\text{DPO-MultiDiffusion}} = &-\log \sigma \Big( \beta \mathbb{E}_{J_{1:T}^w \sim q(J_{1:T}|J_0^w,c), J_{1:T}^l \sim q(J_{1:T}|J_0^l,c)} \\
& \Big[ \log \frac{p_\theta(J_{0:T}^w|c)}{p_{\text{ref}}(J_{0:T}^w|c)} - \log \frac{p_\theta(J_{0:T}^l|c)}{p_{\text{ref}}(J_{0:T}^l|c)} \Big] \Big) \\
= &-\log \sigma \Big( \beta \mathbb{E}_{J_{1:T}^w \sim q(J_{1:T}|J_0^w), J_{1:T}^l \sim q(J_{1:T}|J_0^l)} \\
& \Big[ \sum_{i=1}^{T} \log \frac{p_\theta(J_{t-1}^w|J_t^w)}{p_{\text{ref}}(J_{t-1}^w|J_t^w)} - \log \frac{p_\theta(J_{t-1}^l|J_t^l)}{p_{\text{ref}}(J_{t-1}^l|J_t^l)} \Big] \Big) \\
= &-\log \sigma \Big( \beta \mathbb{E}_{J_{1:T}^w \sim q(J_{1:T}|J_0^w), J_{1:T}^l \sim q(J_{1:T}|J_0^l)} T \mathbb{E}_t \\
& \Big[ \log \frac{p_\theta(J_{t-1}^w|J_t^w)}{p_{\text{ref}}(J_{t-1}^w|J_t^w)} - \log \frac{p_\theta(J_{t-1}^l|J_t^l)}{p_{\text{ref}}(J_{t-1}^l|J_t^l)} \Big] \Big) \\
= &-\log \sigma \Big( \beta T \mathbb{E}_t \mathbb{E}_{J_{t-1,t}^w \sim q(J_{t-1,t}|J_0^w), J_{t-1,t}^l \sim q(J_{t-1,t}|J_0^l)} \\
& \Big[ \log \frac{p_\theta(J_{t-1}^w|J_t^w)}{p_{\text{ref}}(J_{t-1}^w|J_t^w)} - \log \frac{p_\theta(J_{t-1}^l|J_t^l)}{p_{\text{ref}}(J_{t-1}^l|J_t^l)} \Big] \Big) \\
= &-\log \sigma \Big( \beta T \mathbb{E}_{t, J_t^w \sim q(J_t|J_0^w), J_t^l \sim q(J_t|J_0^l)} \\
& \mathbb{E}_{J_{t-1}^w \sim q(J_{t-1}|J_0^w), J_{t-1}^l \sim q(J_{t-1}|J_0^l)} \Big[ \log \frac{p_\theta(J_{t-1}^w|J_t^w)}{p_{\text{ref}}(J_{t-1}^w|J_t^w)} - \log \frac{p_\theta(J_{t-1}^l|J_t^l)}{p_{\text{ref}}(J_{t-1}^l|J_t^l)} \Big] \Big)
\end{aligned}
\tag{34}
$$

By Jenson's inequality, it can be applied:

$$L_{\text{DPO-MultiDiffusion}} \leq$$

$$-\mathbb{E}_{t,J_t^w \sim q(J_t|J_0^w),J_t^l \sim q(J_t|J_0^l)} \log \sigma \Bigg( -\beta T \mathbb{E}_{J_{t-1}^w \sim q(J_{t-1}|J_0^w,J_t^w),J_{t-1}^l \sim q(J_{t-1}|J_0^l,J_t^l)}$$

$$\left[ \log \frac{p_\theta(J_{t-1}^w|J_t^w,c)}{p_{\text{ref}}(J_{t-1}^w|J_t^w,c)} - \log \frac{p_\theta(J_{t-1}^l|J_t^l,c)}{p_{\text{ref}}(J_{t-1}^l|J_t^l,c)} \right] \Bigg)$$

$$= -\mathbb{E}_{t,J_t^w \sim q(J_t|J_0^w),J_t^l \sim q(J_t|J_0^l)} \log \sigma \Bigg( -\beta T \bigg( \tag{35}$$

$$+ \mathbb{D}_{\text{KL}} \big( q(J_{t-1}^w|J_0^w,J_t^w) \,||\, p_\theta(J_{t-1}^w|J_t^w,c) \big)$$

$$- \mathbb{D}_{\text{KL}} \big( q(J_{t-1}^w|J_0^w,J_t^w) \,||\, p_{\text{ref}}(J_{t-1}^w|J_t^w,c) \big)$$

$$- \mathbb{D}_{\text{KL}} \big( q(J_{t-1}^l|J_0^l,J_t^l) \,||\, p_\theta(J_{t-1}^l|J_t^l,c) \big)$$

$$+ \mathbb{D}_{\text{KL}} \big( q(J_{t-1}^l|J_0^l,J_t^l) \,||\, p_{\text{ref}}(J_{t-1}^l|J_t^l,c) \big) \bigg) \Bigg)$$

Thus, the objective function becomes:

$$L(\theta) \approx -\mathbb{E}_{t,J_t^w \sim q(J_t|J_0^w),J_t^l \sim q(J_t|J_0^l)} \log \sigma \Bigg( -\beta T($$

$$\|\mu_t^w - \mu_\theta^w\| - \|\mu_t^w - \mu_{\text{ref}}^w\| - (\|\mu_t^l - \mu_\theta^l\| - \|\mu_t^l - \mu_{\text{ref}}^l\|)) \Bigg)$$

$$= -\mathbb{E}_{t,J_t^w \sim q(J_t|J_0^w),J_t^l \sim q(J_t|J_0^l)} \log \sigma \Bigg( -\beta T( \tag{36}$$

$$\|\mu_t^w - \sum_{i=1}^n F_i^{-1}(\mu_{\theta,i}^w)\| - \|\mu_t^w - \sum_{i=1}^n F_i^{-1}(\mu_{\text{ref},i}^w)\|$$

$$-(\|\mu_t^l - \sum_{i=1}^n F_i^{-1}(\mu_{\theta,i}^l)\| - \|\mu_t^l - \sum_{i=1}^n F_i^{-1}(\mu_{\text{ref},i}^l)\|)) \Bigg)$$

$$L(\theta) = -\mathbb{E}_{t,J_t^w \sim q(J_t|J_0^w),J_t^l \sim q(J_t|J_0^l)} \log \sigma \bigg( (-\beta T \omega(\lambda_t))($$

$$\|\epsilon^w - \sum_{i=1}^n F_i^{-1}(\epsilon_\theta(x_t^w,t))\| - \|\epsilon^w - \sum_{i=1}^n F_i^{-1}(\epsilon_{\text{ref}}(x_t^w,t))\| \tag{37}$$

$$-(\|\epsilon^l - \sum_{i=1}^n F_i^{-1}(\epsilon_\theta(x_t^l,t))\| - \|\epsilon^l - \sum_{i=1}^n F_i^{-1}(\epsilon_{\text{ref}}(x_t^l,t))\|)) \bigg)$$

## A.4 MORE RESULTS

This section presents further qualitative comparisons against baseline methods. These visual results highlight our method's superior ability to maintain global structural coherence while simultaneously rendering finer and more intricate local details. Furthermore, to elucidate the distinct contributions of each component, we present qualitative results from our ablation study in Fig. 8.Beyond this component-level analysis, we also present a diverse portfolio of generated scrolls (Fig. figs. 9 and 10) to underscore the model's stylistic flexibility and its capacity for creative generalization.

## A.5 FAILURE CASES

In this section, we analyze several failure cases (Fig.11) to provide a transparent account of our method's limitations. First, our method can struggle with generating complex semantic concepts, such as the human faces shown in Fig.11a. This is a well-documented challenge in text-to-image

synthesis and highlights that our framework's performance is fundamentally dependent on the capabilities of the underlying base model. Furthermore, while our approach significantly mitigates object repetition, it does not entirely eliminate this artifact, as instances can still occur during the generation of very long scrolls (Figs. figs. 11b and 11c). However, we note that the frequency of such repetitions is substantially reduced compared to baseline methods. Finally, Fig.11d illustrates a semantic error where, despite a globally coherent structure, the model conflates the concepts of a "beach" and "water" from the prompt. This suggests that there is still room for improvement in the model's text-to-image alignment and its ability to interpret complex spatial relationships.

## A.6 SOCIAL IMPACT

The primary impact of our work is to advance the technical capabilities within the field of image generation, specifically for high-aspect-ratio content. Our DRSPO framework addresses the critical and persistent challenge of maintaining global coherence in long-scroll synthesis, enabling the reliable creation of large-scale panoramic and narrative images without common artifacts like object repetition. By providing a method that successfully balances global structure with fine-grained detail, we offer a more practical and effective tool for artists and designers, potentially enhancing creative workflows in digital art, virtual environment design, and sequential media. Furthermore, our application of preference optimization to this task contributes a valuable technique for better aligning generative outputs with human intent, improving the usability and quality of these powerful creative tools.

## A.7 REPRODUCIBILITY STATEMENT

We present a detailed derivation of our method for applying DPO to the MultiDiffusion framework in Sec.3.2, with an extended analysis provided in AppendixA.3. The hyperparameters and settings for our training process are outlined in Sec.5.1.2. Furthermore, Sec.5.1.1 describes our dataset construction pipeline, which includes prompt selection using the Coze platform and the subsequent generation of preference pairs from various base models. Our implementation will be made publicly available on GitHub.

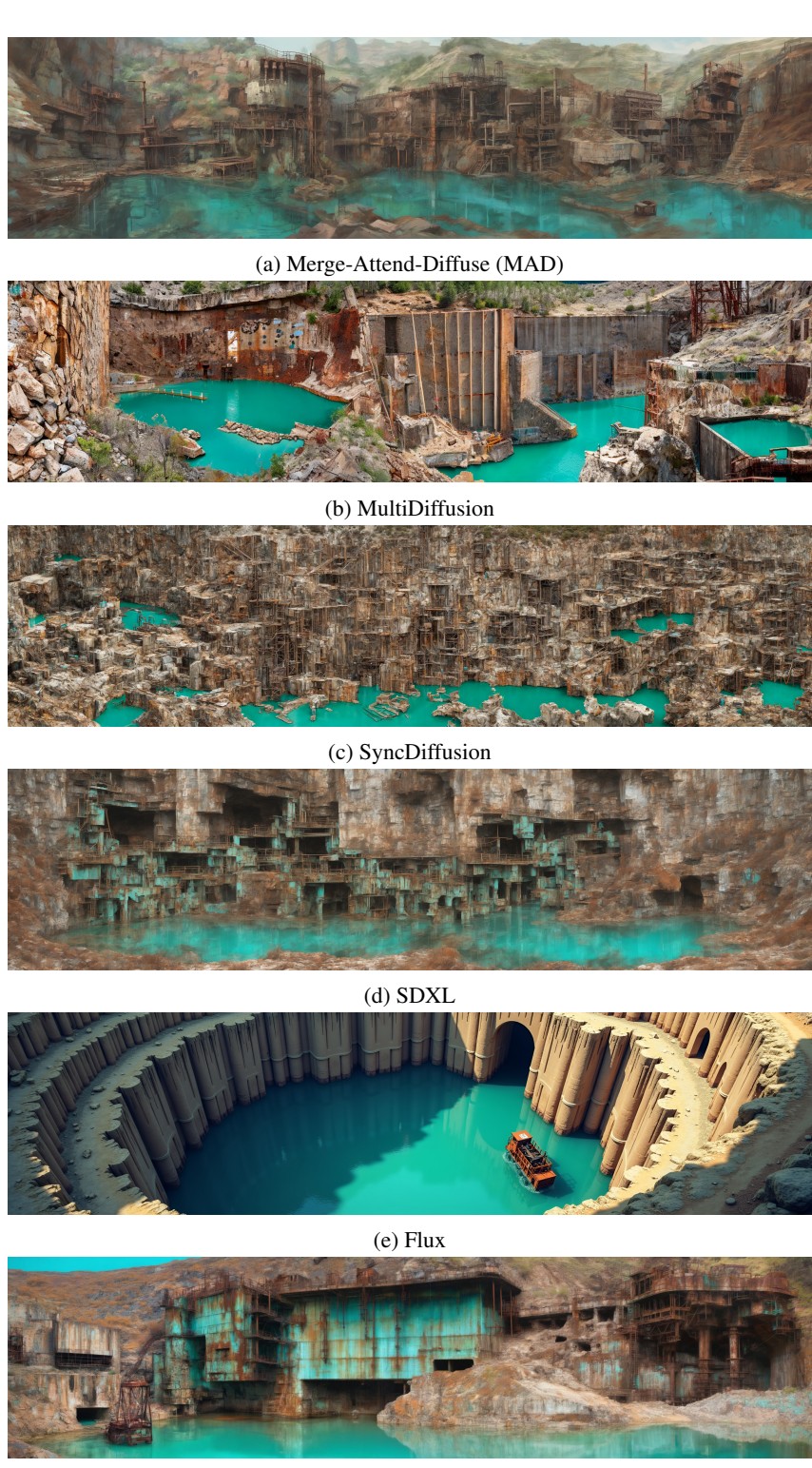

(a) Merge-Attend-Diffuse (MAD)

(b) MultiDiffusion

(c) SyncDiffusion

(d) SDXL

(e) Flux

(f) DRSPO(Ours)

Figure 4: Scrolls(1024×4096) generated using prompts:"Illustrate an old, abandoned quarry. The sheer, tiered rock walls descend to a deep pool of startlingly turquoise water at the bottom. Rusted machinery lies half-submerged, a sign of the industry that once thrived here."

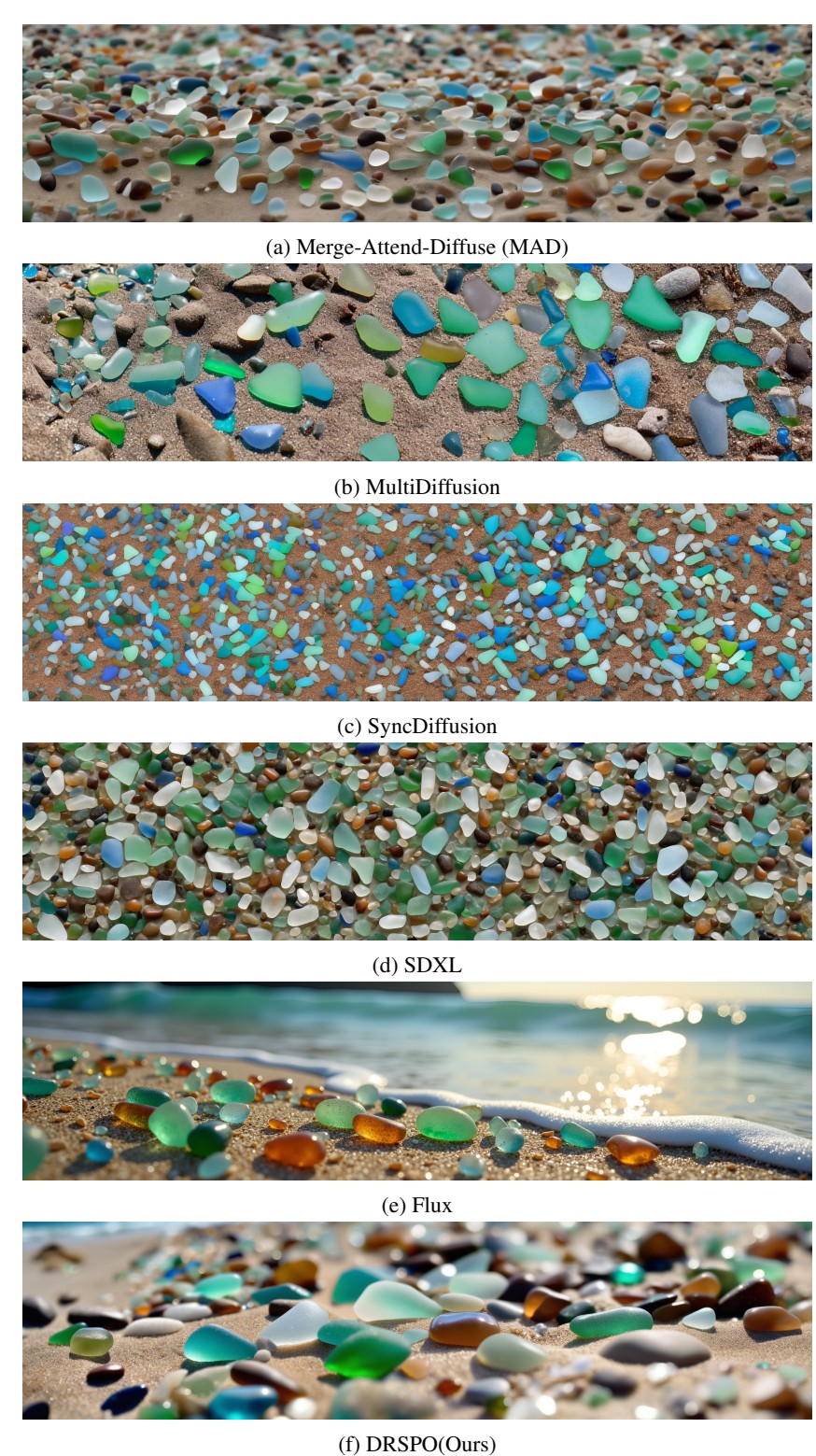

(a) Merge-Attend-Diffuse (MAD)

(b) MultiDiffusion

(c) SyncDiffusion

(d) SDXL

(e) Flux

(f) DRSPO(Ours)

Figure 5: Scrolls(1024×4096) generated using prompts:"Write about a small, hidden beach where the sand is made of smooth, colorful sea glass. The tiny, worn fragments of glass in green, blue, and brown sparkle in the sun like jewels with every wave that washes ashore."

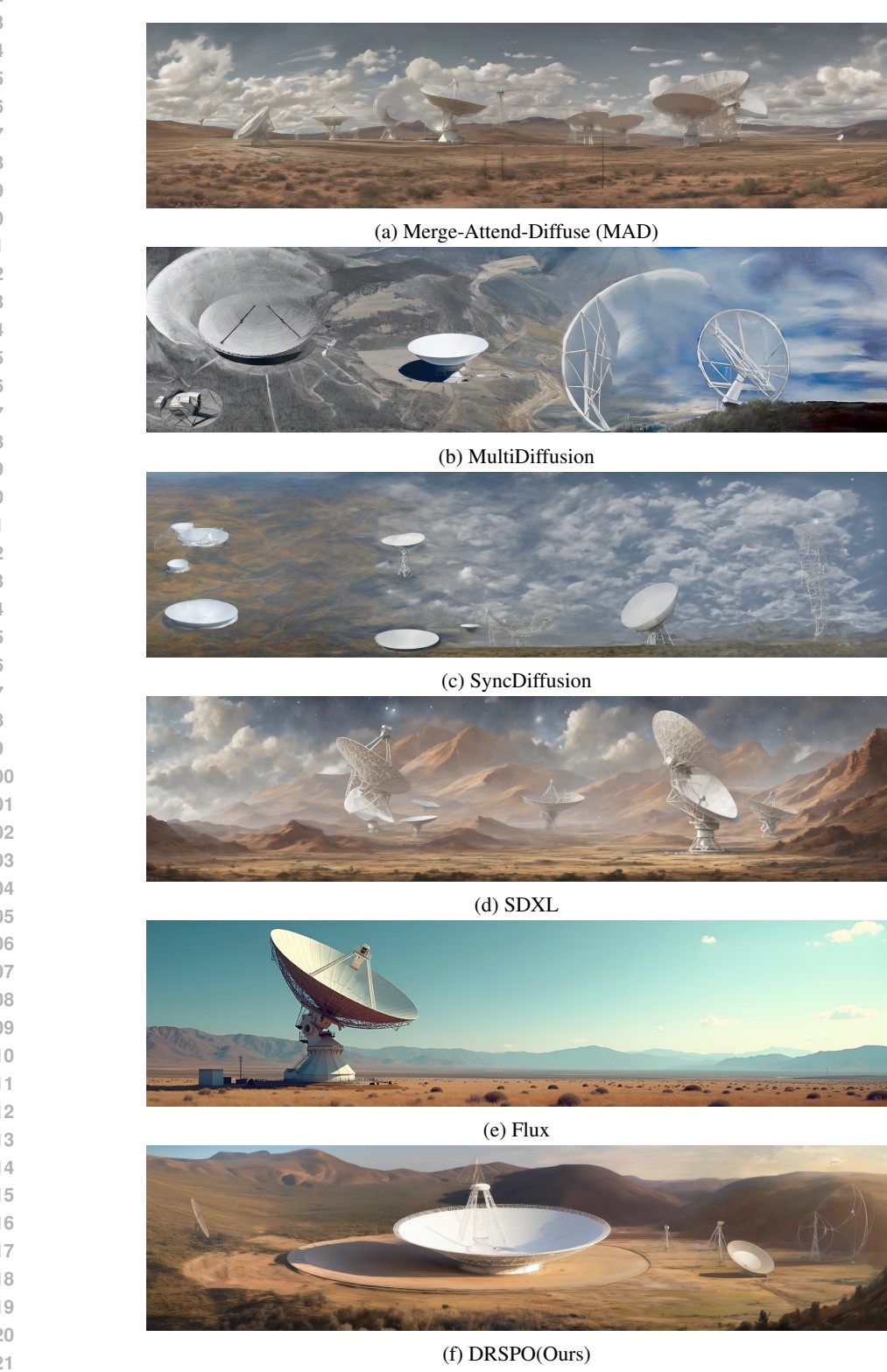

(a) Merge-Attend-Diffuse (MAD)

(b) MultiDiffusion

(c) SyncDiffusion

(d) SDXL

(e) Flux

(f) DRSPO(Ours)

Figure 6: Scrolls(1024×4096) generated using prompts:"Paint a picture of a massive radio telescope dish in a remote, quiet valley. The enormous white dish is pointed towards the sky, silently listening for signals from the depths of the universe, a symbol of human curiosity."

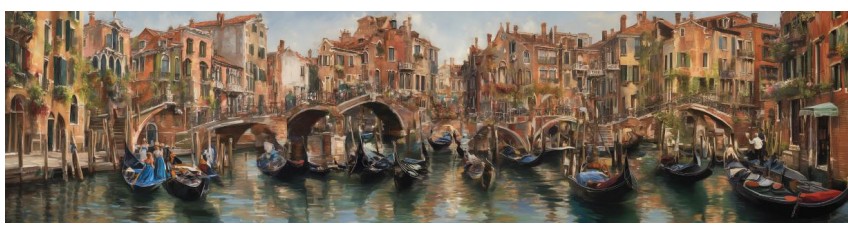

(a) MAD:Aesthetic Score=6.577

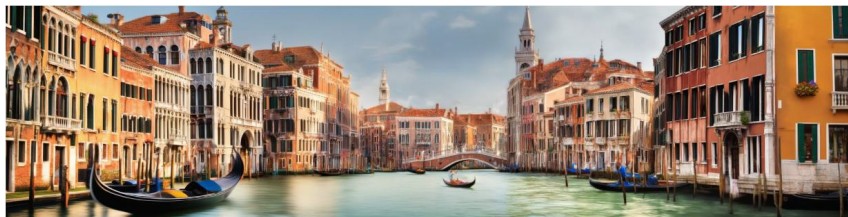

(b) DRSPO(Ours):Aesthetic Score=5.870

Figure 7: Although MAD generates a large number of repetitive and illogical objects, it achieves a higher score on the Aesthetic Score. This suggests that the Aesthetic Score may favor images that generate repetitive objects.

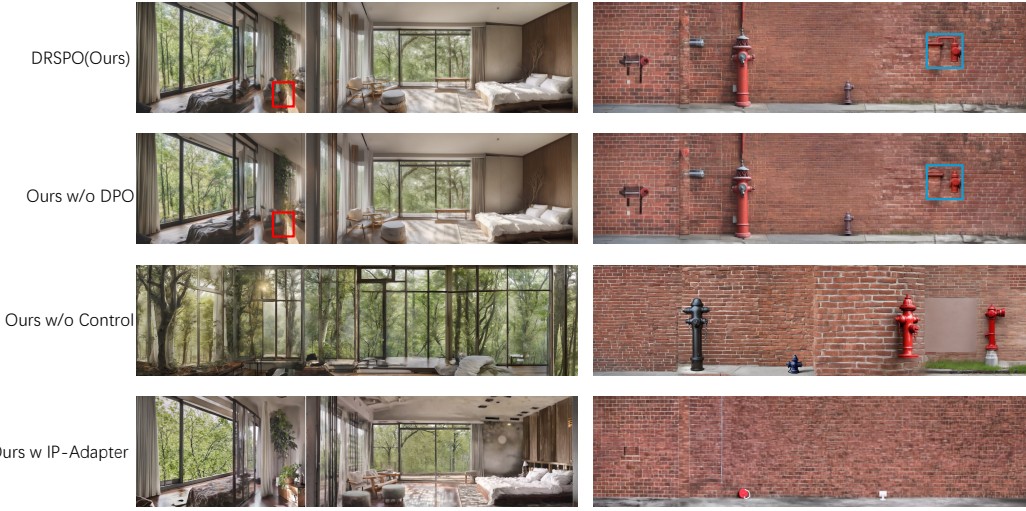

Figure 8: The visual results of our ablation study clearly elucidate the distinct role of each component. Given that the foundational structure is already established by the low-resolution blueprint, the DPO applied in the HR stage is primarily responsible for refining fine-grained details and enhancing local textures. Conversely, the blueprint itself governs the global composition; consequently, its removal leads to a catastrophic collapse of structural integrity, fundamentally altering the entire image. Finally, we observe that integrating an additional IP-Adapter is detrimental, introducing severe visual artifacts such as blurring and the complete omission of objects.

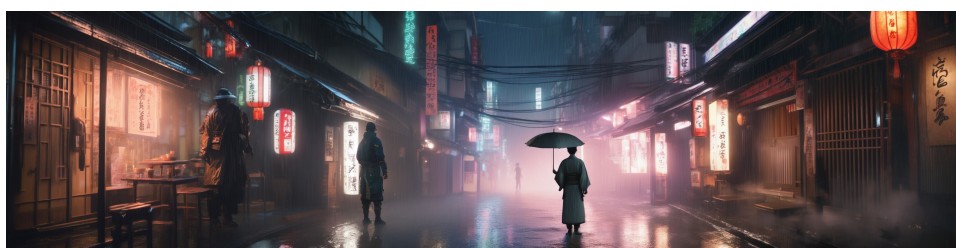

(a) "A lone cyborg ronin standing in a narrow, rain-slicked alley of Neo-Kyoto, steam rising from a ramen stall, the neon glow from flickering holographic advertisements reflecting in the puddles of acid rain, cinematic lighting, volumetric fog, hyperrealistic detail, octane render."

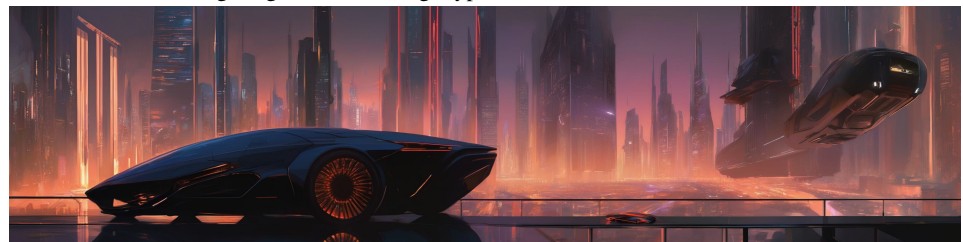

(b) "A sleek, black aerodyne vehicle hovers silently outside the panoramic window of a megacorporation's penthouse, overlooking a sprawling, light-saturated metropolis at night, the interior of the vehicle is dark, hinting at a powerful figure within, wide-angle concept art, style of Syd Mead and Blade Runner 2049."

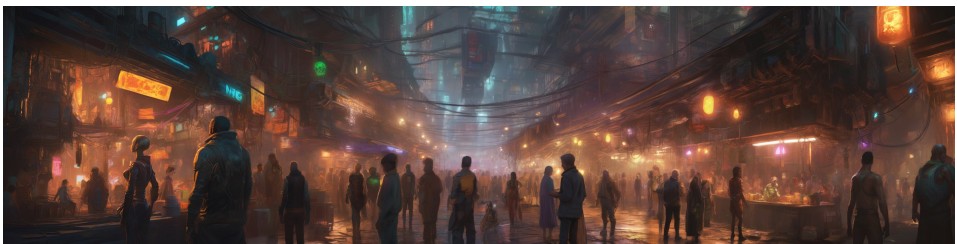

(c) "A bustling, multi-level bazaar in a grimy undercity, crowded with humans, androids, and aliens bartering for black market cybernetics and glowing bio-enhancements, tangled wires and pipes hang overhead, vibrant chaos, dynamic composition, detailed character design, cinematic atmosphere."

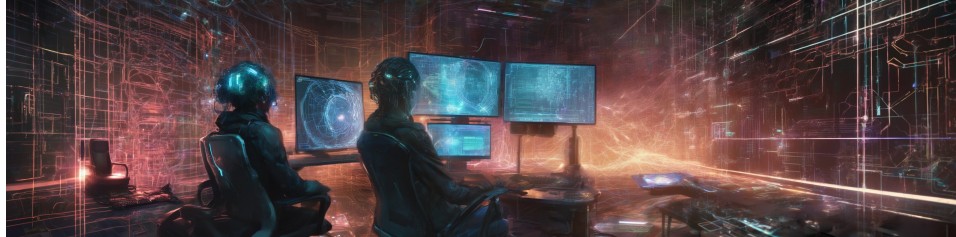

(d) "An elite netrunner slumped in a customized console chair, wires jacked directly into their neural port, surrounded by a chaotic web of holographic screens displaying cascading lines of cryptic code and abstract data-fortress schematics, the only light source is the glow from the monitors, dark and moody, cyberpunk aesthetic."

Figure 9: Scrolls generated by our method using cyberpunk style prompts.

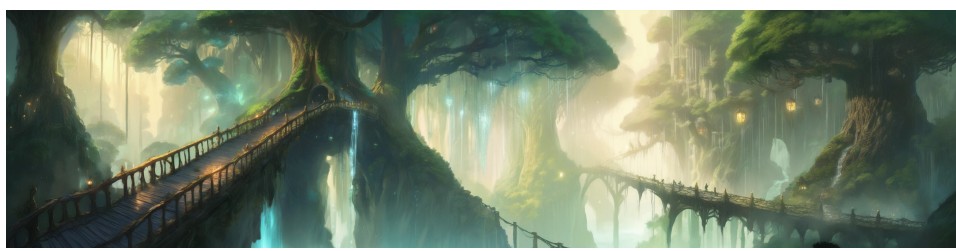

(a) "An ancient, colossal tree whose branches hold an entire elven city, with crystalline bridges connecting glowing lantern-lit homes, waterfalls cascading from the highest limbs into a misty abyss below, fantasy concept art, highly detailed, epic scale, style of Studio Ghibli and Ori and the Blind Forest."

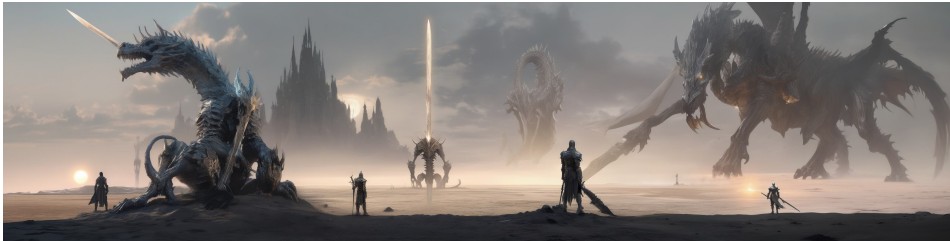

(b) "A lone knight clad in ornate, sun-bleached armor kneels before a dragon's colossal skeleton in a vast, desolate wasteland of black sand, a glowing magical sword plunged into the ground beside him, twin moons illuminating the scene, cinematic, somber atmosphere, hyperrealistic."

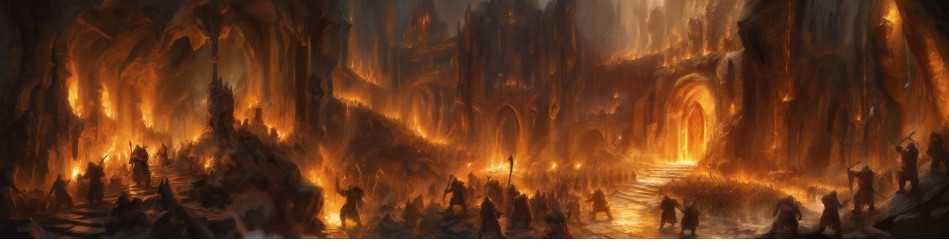

(c) "A bustling underground dwarven forge deep within a mountain, where rivers of molten gold flow in carved channels and master artisans hammer runes of power onto massive war axes, the air thick with sparks and the heat of the earth's core, dramatic lighting, dynamic composition, fantasy art."

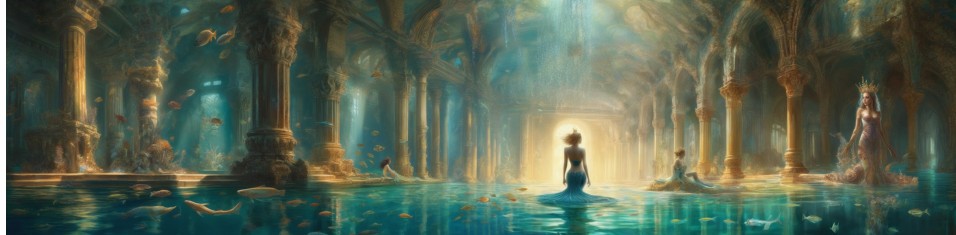

(d) "The grand hall of a forgotten king, now sunken beneath the ocean, with massive coral-encrusted thrones and pillars, schools of bioluminescent fish swim through the ethereal, sun-dappled water, a graceful mermaid queen watches from the shadows, underwater photography, magical realism, highly detailed."

Figure 10: Scrolls generated by our method using fantasy style prompts.

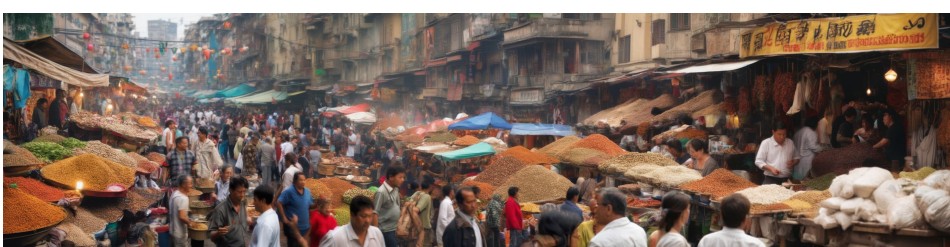

(a) "Write about a crowded open-air market in a foreign city. The air is thick with the smells of spices, street food, and incense, and the sound is a cacophony of vendors shouting, music playing, and people haggling."

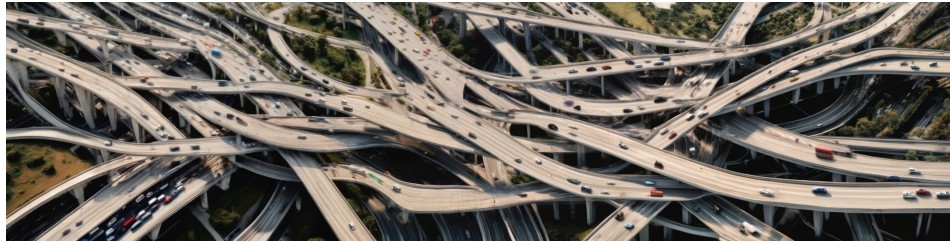

(b) "Describe a vast, multi-level highway interchange, a complex knot of concrete ramps and fly-overs. Cars, trucks, and buses flow in a constant, mesmerizing, and intricate dance of organized chaos."

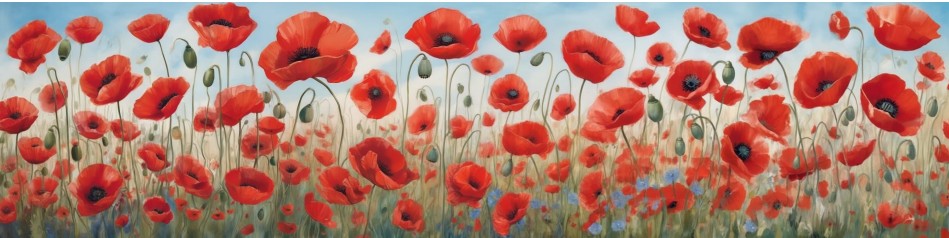

(c) "Illustrate a vast field of poppies, a sea of brilliant red under a clear blue sky. The delicate petals tremble in a gentle breeze, creating a beautiful and poignant scene that is often associated with remembrance."

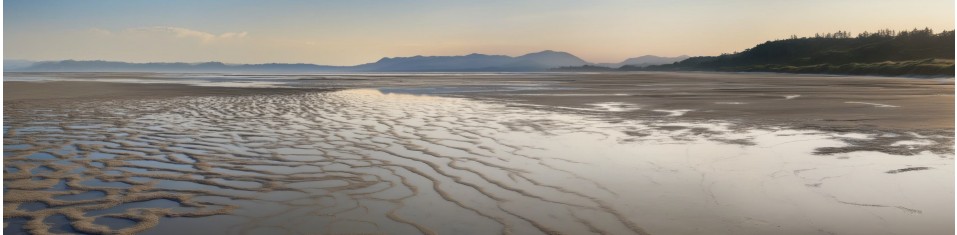

(d) "Depict a vast, empty beach at low tide, stretching for miles. The wet sand reflects the sky like a mirror, and the receding water has left intricate patterns and shallow tidal pools behind."

Figure 11: we present and analyze several failure cases to provide a transparent account of our method's current limitations.

