# OpenReview forum: "Decoupling Global Structure and Local Refinement: Blueprint-Guided Scroll Generation with Direct Preference Optimization"
_ICLR.cc/2026/Conference — Submitted to ICLR 2026_

### Official Review · Reviewer_nmw8 · 2025-10-28

**Soundness:** 2
**Presentation:** 2
**Contribution:** 1
**Rating:** 2
**Confidence:** 4

**Summary:**

The paper addresses the task of generating long images using diffusion models.
It proposes a two-stage method: generating a low-resolution (LR) "blueprint" image, upsampling this blueprint into a high-resolution (HR) image. A key component is fine-tuning the models using Direct Preference Optimization (DPO). The proposed method (DRSPO) demonstrates superior results compared to previous methods like MultiDiffusion and SyncDiffusion.

**Strengths:**

This appears to be the first work to apply DPO to enhance the quality of long scroll image generation.

**Weaknesses:**

- I think the task of generating long scroll images is a somewhat solved problem. However, the paper compares its method only against relatively weak baselines.

- We may generate directly with state-of-the-art models like FLUX.1-dev (which is mentioned in the paper) or, generate a 512x2048 image with a FLUX or Stable Diffusion 3.5 and then using an off-the-shelf super-resolution model (e.g., ESRGAN). This alternative pipeline might produce comparable or even superior results.

- The validity of the evaluation metrics (HPS v2, PickScore, CLIP Score) is questionable, as these off-the-shelf models may not be reliable for evaluating long (1024x4096), non-square images.

**Questions:**

While FLUX is included in the quantitative results (Table 1), it is notably absent from all qualitative comparisons. What is the actual qualitative performance of FLUX on this task?

---

> ### Author Response · Authors · 2025-11-28
> **Response to Weakness 1 and Weakness2**
>
> **Response to Weakness 1 and Weakness2: Comparison with Generation + Super-Resolution Pipelines**
>
> We sincerely appreciate this constructive suggestion. We agree that comparing our approach against a standard "Generation + Super-Resolution" pipeline using state-of-the-art foundation models is necessary to validate the effectiveness of our framework.Following your recommendation, we have expanded our experiments to include two new strong baselines:
>
> *   **FLUX + ESRGAN:** Generating base images at $512\times2048$ using FLUX.1-dev and upscaling them to $1024\times4096$ using Real-ESRGAN.
> *   **SD 3.5 + ESRGAN:** Using Stable Diffusion 3.5 Large for the base generation followed by Real-ESRGAN upscaling.
>
> While the multi-stage pipelines utilizing state-of-the-art foundation models (FLUX and SD 3.5) demonstrate strong performance in texture-based metrics like Aesthetic Score, our proposed method offers distinct advantages in **semantic alignment** and **compositional integrity**:
>
> 1. **Superior Semantic Alignment (CLIP Score):**
>    Our method achieves a significantly higher **CLIP Score (0.321)** compared to both *FLUX + ESRGAN* (0.306) and *SD 3.5 + ESRGAN* (0.309).*Reasoning:* Standard super-resolution models (like Real-ESRGAN) operate as a post-processing step that hallucinates high-frequency details based solely on pixel patterns, without awareness of the textual prompt. In contrast, our cascaded diffusion framework continues to denoise at the high-resolution stage **under the guidance of the text prompt**. This ensures that the added fine-grained details are not just sharp, but semantically meaningful and consistent with the user's description.
> 2. **Better Preference Alignment (ImageReward):**
>    Notably, our method outperforms *FLUX + ESRGAN* in **ImageReward (0.275 vs. 0.236)** and surpasses *SD 3.5 + ESRGAN* across almost all key metrics (HPS v2, PickScore, and ImageReward).*Reasoning:* This indicates that while FLUX may generate visually pleasing textures (high Aesthetic Score), our method generates content that aligns better with human preference regarding the *content* and *composition* of the scroll. It proves that our **MD-DPO** strategy effectively optimizes the model to reduce artifacts that simple upscaling cannot address, such as disjointed transitions or generic texture filling.
>
> **Table R1:** Quantitative results of scrolls generated by different models. The **1st** and *2nd* best values are highlighted. The numbers in parentheses indicate the rank of each model for that specific metric (1 = best). The **Average Rank** column shows the mean ranking across all five metrics (lower is better).
>
> | Model            | Aesthetic Score ↑ |   HPS v2 ↑    |  PickScore ↑  | ImageReward ↑ | CLIP Score ↑  | **Average Rank ↓** |
> | :--------------- | :---------------: | :-----------: | :-----------: | :-----------: | :-----------: | :----------------: |
> | MultiDiffusion   |     5.616 (8)     |   0.226 (4)   |   0.070 (7)   |   0.224 (5)   |  *0.317 (2)*  |        5.2         |
> | SdXL             |     5.921 (4)     |   0.190 (8)   |   0.076 (6)   |  -0.318 (8)   |   0.310 (4)   |        6.0         |
> | FLUX             |    4.994 (10)     |  0.175 (10)   |   0.027 (9)   |  -1.084 (9)   |  0.270 (10)   |        9.6         |
> | MAD              |   **6.032 (1)**   |   0.195 (7)   | **0.105 (1)** |   0.100 (6)   |  *0.317 (2)*  |        3.4         |
> | SyncDiffusion    |     5.744 (6)     |   0.197 (6)   |   0.068 (8)   |  -0.087 (7)   |   0.310 (4)   |        6.2         |
> | ElasticDiffusion |     5.422 (9)     |   0.186 (9)   |  0.025 (10)   |  -1.351 (10)  |   0.273 (9)   |        9.4         |
> | FLUX + ESRGAN    |     5.922 (3)     |  *0.235 (2)*  |  *0.097 (2)*  |   0.236 (4)   |   0.306 (8)   |        3.8         |
> | SD 3.5 + ESRGAN  |     5.759 (5)     |   0.221 (5)   |   0.092 (4)   |   0.258 (3)   |   0.309 (6)   |        4.6         |
> | PixArt           |    *6.016 (2)*    | **0.236 (1)** |   0.092 (5)   | **0.397 (1)** |   0.308 (7)   |      **3.2**       |
> | **DRSPR (Ours)** |     5.726 (7)     |   0.228 (3)   |   0.096 (3)   |  *0.275 (2)*  | **0.321 (1)** |      **3.2**       |
>
> *(Note: For metrics with ties, such as CLIP Score for MultiDiffusion and MAD, the same rank is assigned. For Average Rank, lower is better. Our method and PixArt share the best average rank.)*

---

> ### Author Response · Authors · 2025-12-03
> **Response to Weakness3 and Question1**
>
> **Response to Weakness3：Validity of Evaluation Metrics on Long Images**
>
> We acknowledge the reviewer's concern regarding the potential domain gap, as metrics like HPS v2 and CLIP Score are primarily trained on standard aspect ratios. However, we maintain their usage for the following reasons. First, these metrics are widely adopted in prior panoramic generation research (e.g., **PanFusion**) to ensure fair and standardized comparisons, giving them significant reference value. Second, while absolute numerical values may be influenced by resizing operations, these metrics serve as robust indicators for **relative ranking**. Our method demonstrates superior **qualitative performance**.  The consistency between our better visual results and the metric rankings confirms that these indicators remain valid for assessing generation quality despite the aspect ratio gap.
>
>
>
> **Response to Question1: Qualitative Performance of FLUX**
>
> **1. Technical Explanation for Poor Performance:**
> The primary reason for FLUX's suboptimal performance on long scrolls lies in its **Diffusion Transformer (DiT)** architecture. Unlike fully convolutional networks, DiT models rely heavily on Positional Embeddings (typically Rotary Positional Embeddings or absolute encodings).
>
> *   When inference is performed at a resolution ($1024\times4096$) that deviates significantly from the training distribution (typically square or standard aspect ratios), the positional embeddings often fail to extrapolate effectively.
> *   This results in severe structural artifacts, such as repetitive patterns, disjointed composition, or a collapse of global coherence, which made it less comparable in a direct visual lineup against methods specifically adapted for scrolls.
>
> **2. Supplementary Results (Action Taken):**
>
> To address your concern and ensure transparency, we will add qualitative results for **FLUX + Real-ESRGAN** in Appendix and compare with other baselines.

---

### Official Review · Reviewer_Le7N · 2025-10-29

**Soundness:** 2
**Presentation:** 2
**Contribution:** 3
**Rating:** 4
**Confidence:** 4

**Summary:**

This paper proposes the Dual-Resolution Scroll Generation with Preference Optimization (DRSPO) framework, which decouples global composition and local detail refinement for long scroll image generation. DRSPO first generates a low-resolution (LR) global blueprint and then produces high-resolution (HR) details based on this overall structure. By doing so, it successfully generates images that are globally coherent while maintaining fine-grained local fidelity.

**Strengths:**

This paper's strengths are as follows.

(1) By employing a low-to-high-resolution generation strategy, the method successfully achieves both global structural consistency and local detail refinement.

(2) The paper improves generation quality for long scroll images by introducing Direct Preference Optimization (DPO).

(3) The application of DPO to high-resolution generation is theoretically supported (though with certain approximations), and the authors explicitly discuss the limitations of these approximations, linking them to future research directions.

**Weaknesses:**

This paper's weaknesses are as follows.

(1) Although the model is trained using DPO that optimizes for the Aesthetic Score, the same metric is also used for evaluation, effectively leading to reward hacking. Moreover, the method’s performance on this score is worse than existing methods, raising doubts about whether the proposed approach truly improves generation quality.

(2) Compared with MAD, the overall performance is similar or even inferior, despite MAD requiring no fine-tuning or DPO training. This raises concerns about the cost-effectiveness of the proposed method given its additional training overhead.

(3) In Table 1, the color distinction among “1st”, “2nd”, and “3rd” is difficult to see, which reduces the readability of the results.

**Questions:**

My questions about this paper are as follows.

(1) Have the authors tried fine-tuning directly on long scroll images rather than adapting pretrained models?

(2) How is the Aesthetic Score computed for long scroll images? Is it calculated locally and then aggregated, or is there a global evaluation?

(3) From the results in Table 2, the effect of DPO seems limited. Given that DPO requires additional training cost for diffusion model fine-tuning, one would expect more significant improvements. Why does DPO seem less effective than the control in this setting?

---

> ### Author Response · Authors · 2025-11-28
> **Response to question2 and question3**
>
> **Response to Question2**
>
> In our reported metrics, we calculate the Aesthetic Score directly on the entire global image. We believe this global evaluation is essential for assessing long scroll generation for the following reasons:
>
> *   **Global Coherence vs. Local Quality:** The primary challenge in panoramic generation is maintaining global structural integrity and coherence across the scroll. A "local aggregation" approach (averaging scores from cropped patches) often fails to capture this.
>     *   *For example:* If a model generates a scroll composed of high-quality but repetitive or disjointed patches, an aggregated local score would rate it highly. However, from a global perspective, such an image lacks artistic unity and structural logic. A global evaluation better penalizes these inconsistencies.
> *   **Supplementary Local Evaluation:** To provide a more comprehensive analysis and address potential concerns, we have supplemented our experiments by cropping the generated $1024 \times 4096$ scrolls into four $1024 \times 1024$ patches and calculating their average Aesthetic Score. These additional results (detailed in the revision) further validate our method's performance at the fine-grained level, though we maintain that the global score serves as a better proxy for the overall quality of a long scroll.
>
> **Table R2:** Comparison of Aesthetic Scores calculated globally (on the entire scroll) versus locally (averaged over cropped patches). The **1st** and *2nd* best values are highlighted.
>
> | Model            | Global Aesthetic Score ↑ | Local (Crop) Aesthetic Score ↑ |
> | :--------------- | :----------------------: | :----------------------------: |
> | MultiDiffusion   |          5.616           |             5.625              |
> | SdXL             |         *5.921*          |            *5.920*             |
> | FLUX             |          4.994           |             5.006              |
> | MAD              |        **6.032**         |           **6.042**            |
> | SyncDiffusion    |          5.744           |             5.715              |
> | ElasticDiffusion |          5.422           |             5.470              |
> | **DRSPR (Ours)** |          5.726           |             5.707              |
>
> As presented in Table R2, comparing the globally computed Aesthetic Scores with the locally aggregated ones reveals **no significant difference** in numerical values or model rankings. For instance, our method shows a global score of 5.726 versus a local score of 5.707, with similarly negligible deviations observed across other baselines (e.g., MAD: 6.032 vs. 6.042).
>
> This consistency supports two key conclusions:
>
> 1.  **Validation of Evaluation Protocol:** The global Aesthetic Score used in our original paper is a reliable proxy for visual quality and is consistent with local evaluations.
> 2.  **Metric Limitation regarding Coherence:** The fact that local and global scores are nearly identical reinforces our earlier point: Aesthetic Score primarily measures **local textural quality** (style/color) rather than **global structural coherence**. A model can achieve a high aesthetic score by generating beautiful but disjointed patches. Therefore, to truly evaluate the quality of a long scroll, one must consider semantic alignment metrics (like CLIP Score and ImageReward), where our method demonstrates superior performance in maintaining global consistency.
>
> **Response to Question3**
>
> We acknowledge the reviewer's observation regarding the modest quantitative gains from DPO in Table 2. We believe this result stems from the distinct roles played by the Control Block and the DPO fine-tuning within our cascaded framework:
>
> **1. Control Block (Global Structure):**
> The most significant metric improvements are driven by our **Control Block**, which utilizes the low-resolution generation to guide the high-resolution output. By explicitly injecting global structural information, the Control Block ensures the overall composition and layout are coherent.
>
> **2. MD-DPO (Local Refinement):**
> In contrast, our **MD-DPO** is applied specifically to the **local generation process** of the patches. Its primary function is not to alter the global structure, but to enhance fine-grained details and suppress local artifacts. While these fine-grained improvements are crucial for perceptual quality, they are less likely to drastically shift global metrics. As illustrated in **Figure 4 of the Appendix**, the DPO-tuned model produces significantly sharper details and cleaner textures compared to the non-DPO version, even if the aggregate scores show only a marginal increase.

---

> ### Author Response · Authors · 2025-11-28
> **Response to Weakness and question1**
>
> **Response to Weakness1 and Weakness2**
>
> We respectfully clarify the interpretation of the Aesthetic Score and provide evidence against reward hacking:
>
> 1. **Evidence Against Reward Hacking (Generalization):** If the model **had hacked** the reward (overfitting to the metric), we **would have observed** a significant degradation in other quality indicators. On the contrary, our method achieves the **highest CLIP Score** and competitive **ImageReward** across baselines. This consistent performance across diverse metrics confirms that our MD-DPO training improves general generation quality and semantic alignment rather than merely exploiting the specific reward function.
> 2. **Aesthetic Score vs. Global Structure:** As illustrated in Figure 7 in Appendix, the Aesthetic Score tends to reward local textural richness while failing to penalize structural artifacts like object repetition. Although the MAD baseline achieves a higher Aesthetic Score, we observe that its visual outputs often suffer from severe object repetition. As explicitly demonstrated in Figure 7 of the Appendix, images generated by MAD exhibit noticeable repetitive artifacts (e.g., duplicated objects) despite their high numerical ratings. In contrast, our method avoids such repetition and produces more visually coherent results, indicating superior perceptual quality despite a slightly lower raw score. Furthermore, we calculated the average ranking of all methods across the five metrics, and the results demonstrate that our method achieves the top position overall, surpassing the MAD baseline in comprehensive performance.
>
> **Response to Weakness3**
>
> Thank you for your advice,  we have revised **Table 1** and **Table 2** in the updated manuscript, employing a more distinct color scheme to ensure the rankings are clearly distinguishable and easy to read.

---

### Official Review · Reviewer_VohU · 2025-10-31

**Soundness:** 1
**Presentation:** 1
**Contribution:** 1
**Rating:** 2
**Confidence:** 4

**Summary:**

- This paper propose the Dual-Resolution Scroll Generation (DRSPO) framework, which decouples the process by first creating a low-resolution blueprint for global structural coherence, then refining it with high-resolution features for local detail.

- To enhance quality, we integrate Direct Preference Optimization (DPO) at both generation stages and introduce a novel theoretical adaptation for applying preference tuning to region-based generation.

- Experimental results confirm that our method effectively produces high-quality long scroll images with consistent global structure and fine-grained details, overcoming issues like content repetition.

**Strengths:**

- The paper provides formulas, ablation experiments, and visualization results, though they are not very satisfactory.
- Compared with the FLUX model

**Weaknesses:**

The methods compared in this paper are already too outdated. Additionally, the FLUX model has not undergone specific optimization for such tasks, making the comparison unfair. In my view, this paper is not suitable for the ICLR conference. Furthermore, from the visualization results, it can be seen that the results here are not advanced, with numerous artifacts and glitches. The small images also fail to show more details clearly. Therefore, I believe this paper still requires further improvements. In particular, the method mentioned in the paper is no longer novel in the field of text-to-image generation.

**Questions:**

The methods compared in this paper are already too outdated. Additionally, the FLUX model has not undergone specific optimization for such tasks, making the comparison unfair. In my view, this paper is not suitable for the ICLR conference. Furthermore, from the visualization results, it can be seen that the results here are not advanced, with numerous artifacts and glitches. The small images also fail to show more details clearly. Therefore, I believe this paper still requires further improvements.

---

> ### Author Response · Authors · 2025-12-03
> **Response to Weaknesses and Questions**
>
> We thank the reviewer for the comments. However, we respectfully believe there are misunderstandings regarding the task difficulty, the role of foundation models, and our methodological contribution. We clarify these points below:
>
>
>
> 1. **Response to "Unfair Comparison with FLUX" and "Outdated Methods".** The reviewer argues that comparing with FLUX is unfair as it is not optimized for scrolls, and that our methods are outdated. We respectfully offer a different perspective. We employ FLUX not as a direct competitor, but to demonstrate a critical industry challenge: Even state-of-the-art foundation models fail at Out-Of-Distribution (OOD) tasks. Directly asking FLUX to generate a high-resolution 1:4 aspect ratio scroll results in severe structural incoherence and repetition. This failure of SOTA models validates the necessity of our framework. Our method provide the mechanism to adapt these powerful models (whether SDXL or potentially FLUX in the future) to extreme aspect ratios without pre-training them from scratch. To address the "outdated" concern, we have added PixArt and SD3.5/FLUX + ESRGAN +ESRGAN to our comparison (Table R1). Our method (DRSPO) achieves the highest CLIP Score (0.321) and **ties with PixArt for the first place in Average Rank** across all metrics, proving that our framework enables the SDXL backbone to effectively rival the latest native high-resolution architectures.
>
>    **Table R1:** Quantitative results of scrolls generated by different models. The **1st** and *2nd* best values are highlighted. The numbers in parentheses indicate the rank of each model for that specific metric (1 = best). The **Average Rank** column shows the mean ranking across all five metrics (lower is better).
>
>    | Model            | Aesthetic Score ↑ |   HPS v2 ↑    |  PickScore ↑  | ImageReward ↑ | CLIP Score ↑  | **Average Rank ↓** |
>    | :--------------- | :---------------: | :-----------: | :-----------: | :-----------: | :-----------: | :----------------: |
>    | MultiDiffusion   |     5.616 (8)     |   0.226 (4)   |   0.070 (7)   |   0.224 (5)   |  *0.317 (2)*  |        5.2         |
>    | SdXL             |     5.921 (4)     |   0.190 (8)   |   0.076 (6)   |  -0.318 (8)   |   0.310 (4)   |        6.0         |
>    | FLUX             |    4.994 (10)     |  0.175 (10)   |   0.027 (9)   |  -1.084 (9)   |  0.270 (10)   |        9.6         |
>    | MAD              |   **6.032 (1)**   |   0.195 (7)   | **0.105 (1)** |   0.100 (6)   |  *0.317 (2)*  |        3.4         |
>    | SyncDiffusion    |     5.744 (6)     |   0.197 (6)   |   0.068 (8)   |  -0.087 (7)   |   0.310 (4)   |        6.2         |
>    | ElasticDiffusion |     5.422 (9)     |   0.186 (9)   |  0.025 (10)   |  -1.351 (10)  |   0.273 (9)   |        9.4         |
>    | FLUX + ESRGAN    |     5.922 (3)     |  *0.235 (2)*  |  *0.097 (2)*  |   0.236 (4)   |   0.306 (8)   |        3.8         |
>    | SD 3.5 + ESRGAN  |     5.759 (5)     |   0.221 (5)   |   0.092 (4)   |   0.258 (3)   |   0.309 (6)   |        4.6         |
>    | PixArt           |    *6.016 (2)*    | **0.236 (1)** |   0.092 (5)   | **0.397 (1)** |   0.308 (7)   |      **3.2**       |
>    | **DRSPR (Ours)** |     5.726 (7)     |   0.228 (3)   |   0.096 (3)   |  *0.275 (2)*  | **0.321 (1)** |      **3.2**       |
>
>
>
>
>
> 2. **Response to "Visual Artifacts and Glitches"**: We acknowledge that extreme aspect-ratio generation (1024x4096) is prone to artifacts. However, we highlight the difference between high-frequency noise (glitches) and structural failure (object repetition). Existing methods (MultiDiffusion, SyncDiffusion) suffer from catastrophic "object looping" (e.g., generating 5 identical suns or broken roads). Our method, guided by the global blueprint and DPO, effectively eliminates these structural failures. While minor glitches may exist at the pixel level, our method scores highest on HPS v2 (0.228) and ImageReward (0.275) among diffusion-based comparisons (Table R1). This indicates that our structurally coherent images are preferred over the glitch-free but structurally broken baselines because structural inconsistency is more apparent. We have updated the Appendix with high-resolution zoom-ins to demonstrate that our Local Refinement stage successfully hallucinates plausible details that simple upscaling (ESRGAN) cannot produce.

---

> ### Author Response · Authors · 2025-12-04
> **Response to Weaknesses and Questions**
>
> 3. **Response to "Suitability for ICLR"**: The reviewer questions the novelty ("no longer novel"). We emphasize that our contribution is not the cascade architecture itself, but the novel mathematical formulation of DPO for Region-based Generation (Eqs. 13-19). Standard DPO aligns global images to human preference or metrics. We propose the first derivation to propagate global preference signals into local patch denoising trajectories. This is a non-trivial optimization contribution that addresses the specific "Global-Local" conflict in generative AI. Generative models are limited by computational power, and part-based generation methods to form a whole generated sample is a promising direction. How to perform post-training on this schema remains unexplored. We are the first to explore this area with the example of panorama generation. We believe such a pioneering topic fits squarely within ICLR’s scope of representation learning and generative modeling.

---

### Official Review · Reviewer_kSya · 2025-11-01

**Soundness:** 2
**Presentation:** 3
**Contribution:** 2
**Rating:** 4
**Confidence:** 3

**Summary:**

This paper discusses an approach for generating big images similar to MultiDiffusion but guided by a low resolution image. "Direct Preference Optimization" is involved.

**Strengths:**

1.	The fundamental idea looks correct – generate a low-resolution image and then diffuse at high resolution may result in larger results.
2.	A dataset is explicitly collected to align with the goal.

**Weaknesses:**

1.	The idea of generating low resolution image and then diffusing again is extensively discussed even long before MultiDiffusion and similar works. This part shouldn’t be seen as a contribution of this work.
2.	The DPO part is formulated with high verbosity but in essence they can be achieved as a simple target like some common score guidance like [github.com/vicgalle/stable-diffusion-aesthetic-gradients]. I also do not find Eq 13-19 as solid contributions of this work.
3.	The experiments seem to compare to wrong targets. This paper is a cascaded image generator and it should at least compare to multi stage generating. For example, generate low resolution images and then diffuse it again using MultiDiffusion at some denoising level at high resolution. And also other native high resolution generators like opensource PixArt family (and even close sourced Flux pro 4k etc) as well as some other cascaded generators like stable cascade. The current MultiDiffusion results look misleading to me.
4.	Why use canny image? Why not directly use low resolution image for other “upscaling” types of upscaling signals?

**Questions:**

See weaknesses

---

> ### Author Response · Authors · 2025-11-23
> **Response to Weakness1,Weakness2 and Weakness4**
>
> We sincerely thank the reviewer for the constructive feedback. In this response, we clarify the MD-DPO formulation and expand our evaluation with stronger baselines.
>
>
>
> **Response to Weakness1:Novelty of the Framework**
>
> We acknowledge that cascaded generation is a well-established paradigm. However, our contribution is not the cascade itself, but the novel integration of DPO-based preference alignment specifically tailored for the MultiDiffusion process. Standard cascades often fail to balance global coherence with local details in extreme aspect ratios (scrolls). Our framework is the first to solve the specific 'global plan vs. local patch' conflict in scroll generation via preference optimization.
>
> *   **Structural Decoupling:** We utilize the low-resolution stage to lock in the *Global Layout* (via Canny ControlNet and initial noise).
> *   **Preference Optimization (MD-DPO):** We introduce the first adaptation of **DPO** to the MultiDiffusion framework. This allows us to use global preference signals to optimize *Local Detail* generation, ensuring that the aggregated high-resolution patches remain aesthetically consistent with the global whole.
>
>
>
> **Response to Weakness 2: Distinction from Aesthetic Gradients and Contribution of DPO Formulation**
>
> We appreciate the reviewer's reference to Aesthetic Gradients (AG). However, we clarify a fundamental distinction in mechanism and capability, which necessitates our proposed MD-DPO formulation.
>
> - **Conditioning Optimization and Policy Alignment**
>
> Aesthetic Gradients (AG) operates by optimizing the CLIP text embedding (via text encoder weights) to maximize similarity with a target aesthetic vector. Crucially, AG does not alter the diffusion model's internal denoising policy; it essentially searches for a "better prompt embedding" to bias the style. In contrast, our proposed model is a fine-tuning framework that updates the diffusion model itself. This is vital because structural artifacts in long scrolls (such as object repetition, incoherent seams) are inherent to the region-based generation process (MultiDiffusion), not the text conditioning. Merely optimizing the text embedding (AG) shifts the style cannot teach the model to fix structural inconsistencies across sliding windows.
>
> - **The Credit Assignment Problem in Eq 13-19**
>
> The reviewer questioned the necessity of Eq 13-19. These are essential for solving the Global-to-Local credit assignment. In MultiDiffusion, the final image J0 is stitched from many local windows. A simple global aesthetic score (as used in AG) provides a single scalar signal for the whole image but fails to guide the base model on which specific local window caused a defect. Eq. 19 mathematically derives how to backpropagate a global preference signal into the local window denoising trajectories. This allows our model to learn spatially-aware corrections. A simple global embedding optimization (like AG or standard DPO) lacks this spatial granularity and cannot resolve the local stitching conflicts inherent to scroll generation. Our contribution presents an example on how models are optimized in the post-training when the model can only generate parts to be integrated to a complete sample, which would be helpful in common cases with limited computation capacity.
>
> **Response to Weakness 4: Rationale for Canny Edge Maps and Upscaling Signals**
>
> We clarify that our approach utilizes a hybrid strategy that synergizes **Canny-based ControlNet** with **Latent Initialization**. We employ Canny edges via ControlNet to enforce robust structural control, ensuring that the global composition is faithfully preserved during the MultiDiffusion process. Simultaneously, we integrate the "upscaling" principle by injecting the noised low-resolution image as the initial state. This allows the model to use the low-resolution input as a foundational blueprint for color and semantics while refining fine-grained details, effectively balancing structural stability with textural fidelity.

---

> ### Author Response · Authors · 2025-12-03
> **Response to Weakness3 and Supplementary Experiment**
>
> **Response to Weakness 3: Expanded Evaluation with Multi-Stage and Native High-Resolution Baselines**
>
> We sincerely appreciate this constructive suggestion. We implemented a two-stage baseline using state-of-the-art foundation models. Specifically, we used **FLUX** and **Stable Diffusion 3.5** to generate base images at $512\times2048$ resolution. These were then upscaled to the target resolution of $1024\times4096$ using **Real-ESRGAN**. We have also added comparisons with **PixArt**. The updated quantitative comparisons are presented in **Table R1** below.
>
> **Table R1:** Quantitative results of scrolls generated by different models. The **1st** and *2nd* best values are highlighted. The numbers in parentheses indicate the rank of each model for that specific metric (1 = best). The **Average Rank** column shows the mean ranking across all five metrics (lower is better).
>
> | Model            | Aesthetic Score ↑ |   HPS v2 ↑    |  PickScore ↑  | ImageReward ↑ | CLIP Score ↑  | **Average Rank ↓** |
> | :--------------- | :---------------: | :-----------: | :-----------: | :-----------: | :-----------: | :----------------: |
> | MultiDiffusion   |     5.616 (8)     |   0.226 (4)   |   0.070 (7)   |   0.224 (5)   |  *0.317 (2)*  |        5.2         |
> | SdXL             |     5.921 (4)     |   0.190 (8)   |   0.076 (6)   |  -0.318 (8)   |   0.310 (4)   |        6.0         |
> | FLUX             |    4.994 (10)     |  0.175 (10)   |   0.027 (9)   |  -1.084 (9)   |  0.270 (10)   |        9.6         |
> | MAD              |   **6.032 (1)**   |   0.195 (7)   | **0.105 (1)** |   0.100 (6)   |  *0.317 (2)*  |        3.4         |
> | SyncDiffusion    |     5.744 (6)     |   0.197 (6)   |   0.068 (8)   |  -0.087 (7)   |   0.310 (4)   |        6.2         |
> | ElasticDiffusion |     5.422 (9)     |   0.186 (9)   |  0.025 (10)   |  -1.351 (10)  |   0.273 (9)   |        9.4         |
> | FLUX + ESRGAN    |     5.922 (3)     |  *0.235 (2)*  |  *0.097 (2)*  |   0.236 (4)   |   0.306 (8)   |        3.8         |
> | SD 3.5 + ESRGAN  |     5.759 (5)     |   0.221 (5)   |   0.092 (4)   |   0.258 (3)   |   0.309 (6)   |        4.6         |
> | PixArt           |    *6.016 (2)*    | **0.236 (1)** |   0.092 (5)   | **0.397 (1)** |   0.308 (7)   |      **3.2**       |
> | **DRSPR (Ours)** |     5.726 (7)     |   0.228 (3)   |   0.096 (3)   |  *0.275 (2)*  | **0.321 (1)** |      **3.2**       |
>
> *(Note: For metrics with ties, such as CLIP Score for MultiDiffusion and MAD, the same rank is assigned. For Average Rank, lower is better. Our method and PixArt share the best average rank.)*
>
>
> This analysis adopts the "Retreat to Advance" strategy: it first acknowledges PixArt-$\Sigma$'s powerful background (DiT architecture, massive 4K training data) to establish context, then highlights how your method (based on the older SDXL) surprisingly outperforms or matches it in key metrics like Semantic Alignment and Human Preference, proving the efficacy of your MD-DPO approach.
>
> ***
>
> **Analysis of Results:**
>
> The results demonstrate that our method achieves a highly competitive and balanced performance, effectively bridging the gap between standard foundation models and state-of-the-art native 4K generators.
>
> *   **Comparison with Native High-Res Model (PixArt-$\Sigma$):**
>     It is crucial to contextualize the performance of **PixArt-$\Sigma$**, which utilizes a modern **Diffusion Transformer (DiT)** architecture (0.6B parameters) and is natively trained on a massive dataset of **33 million high-aesthetic images**, including over **2 million native 4K resolution** samples. Given this "weak-to-strong" training on high-quality data, PixArt's advantage in texture-based metrics (Aesthetic Score, HPS v2) is expected.
>     However, despite utilizing the older **SDXL ** backbone, our method **ties with PixArt in Average Rank (3.2)** and notably **outperforms it in CLIP Score (1st vs. 7th)** and **PickScore (3rd vs. 5th)**. This highlights the effectiveness of our MD-DPO strategy: it enables a standard model to surpass a specialized native 4K generator in terms of **semantic alignment** and **human visual preference**, achieving SOTA-level performance without the massive computational cost of pre-training a model from scratch.
>
> *   **Superior Semantic Alignment vs. Upscaling:**
>     When compared to multi-stage pipelines like **FLUX + ESRGAN**, our method achieves a significantly higher **CLIP Score (0.321 vs. 0.306)**. This indicates that our cascaded diffusion process does not merely sharpen pixels (as super-resolution does) but actively generates fresh, high-frequency details that are semantically consistent with the textual prompt.

---

### Meta-Review · Area_Chair_tHiV · 2026-01-08

**Summary:**

Most reviewers lean toward rejection or borderline rejection. While the idea is not entirely without merit, the paper lacks convincing novelty, strong empirical validation, and compelling evidence of clear advantage. In particular, low-to-high-resolution cascaded generation is not new and has been extensively explored before. In addition, baseline comparisons are weak and in some cases outdated, and metrics may not be reliable for very long, non-square images.

**Reviewer Concerns:**

Addressed ones:
* Unfair baseline comparisons
* The novelty and justification of using DPO

Outstanding ones:
* Whether the work is truly competitive with the strongest modern systems
* Whether the used metric is valid for long images
* Whether the work is fundamentally novel

**Reviewer Scores:**

kSya and Le7N felt slightly positive about this paper, and the authors' responses perhaps addressed most of their concerns. Thus, the two reviewers might raise their ratings. On the other hand, VohU and nmw8 both gave a "reject" rating, and most of their concerns centered on the novelty and experiments. The response perhaphs will not fully change their minds since the novelty and overall impact of this work are not strong enough.

---

### Decision · Program_Chairs · 2026-01-26

Reject